# Conformational interconversion of MLKL and disengagement from RIPK3 precede cell death by necroptosis

Sarah E. Garnish[1,2,8], Yanxiang Meng [1,2,8], Akiko Koide[3,4,8], Jarrod J. Sandow [1,2], Eric Denbaum[3], Annette V. Jacobsen [1,2], Wayland Yeung[5], Andre L. Samson [1,2], Christopher R. Horne [1,2], Cheree Fitzgibbon[1], Samuel N. Young[1], Phoebe P. C. Smith [1], Andrew I. Webb [1,2], Emma J. Petrie[1,2], Joanne M. Hildebrand [1,2], Natarajan Kannan[5,6], Peter E. Czabotar [1,2], Shohei Koide [3,7,9 ✉] & James M. Murphy [1,2,9 ✉]

Phosphorylation of the MLKL pseudokinase by the RIPK3 kinase leads to MLKL oligomerization, translocation to, and permeabilization of, the plasma membrane to induce necroptotic cell death. The precise choreography of MLKL activation remains incompletely understood. Here, we report Monobodies, synthetic binding proteins, that bind the pseudokinase domain of MLKL within human cells and their crystal structures in complex with the human MLKL pseudokinase domain. While Monobody-32 constitutively binds the MLKL hinge region, Monobody-27 binds MLKL via an epitope that overlaps the RIPK3 binding site and is only exposed after phosphorylated MLKL disengages from RIPK3 following necroptotic stimulation. The crystal structures identified two distinct conformations of the MLKL pseudokinase domain, supporting the idea that a conformational transition accompanies MLKL disengagement from RIPK3. These studies provide further evidence that MLKL undergoes a large conformational change upon activation, and identify MLKL disengagement from RIPK3 as a key regulatory step in the necroptosis pathway.

[1] Walter and Eliza Hall Institute of Medical Research, Parkville, VIC, Australia. [2] Department of Medical Biology, University of Melbourne, Parkville, VIC, Australia. [3] Perlmutter Cancer Center, New York University Langone Health, New York, NY, USA. [4] Department of Medicine, New York University Grossman School of Medicine, New York, NY, USA. [5] Institute of Bioinformatics, University of Georgia, Athens, GA, USA. [6] Department of Biochemistry and Molecular Biology, University of Georgia, Athens, GA, USA. [7] Department of Biochemistry and Molecular Pharmacology, New York University Grossman School of Medicine, New York, NY, USA. [8] These authors contributed equally: Sarah E. Garnish, Yanxiang Meng, Akiko Koide. [9] These authors jointly supervised this work: Shohei Koide, James M. Murphy. ✉email: Shohei.Koide@nyulangone.org; jamesm@wehi.edu.au

**N**ecroptosis is a lytic form of programmed cell death[1–6] that is dysregulated in multiple human pathologies, including inflammatory diseases[7–9], inflammatory bowel disease[10], and ischemia-reperfusion injury[11–13]. The necroptotic pathway is considered to have originated as a host defense mechanism against invading pathogens[14–18]. Necroptosis results in the compromise of plasma membrane integrity and the release of inflammatory cellular contents, including damage-associated molecular patterns (DAMPs). Necroptosis can be triggered by a number of stimuli, including tumor necrosis factor (TNF) receptor superfamily members, Toll-like receptors, and nucleic acid sensors[1,19,20]. Among these, the necroptotic signaling stimulated by the binding of TNF to TNF receptor 1 is the most well-studied. In scenarios where receptor-interacting serine/threonine protein kinase (RIPK)-1 is unable to be ubiquitylated by the cellular inhibitors of apoptosis proteins (cIAPs) E3 ubiquitin ligase family, and the apoptotic proteolytic enzyme, Caspase-8, is inhibited, a high molecular weight complex termed the "necrosome"[21,22] is formed. Within the necrosome, RIPK1 and RIPK3 form a heterooligomeric complex, and RIPK3 is activated by autophosphorylation[3,5]. The precise molecular details of what follows RIPK3 activation are currently not well understood, but it is proposed that RIPK3 recruits and phosphorylates the terminal effector protein, mixed lineage kinase domain-like (MLKL)[4,6,23–26]. MLKL comprises an N-terminal four-helix bundle (4HB) domain, which is responsible for membrane disruption and cell death, a two-helix connector termed the brace helices, and a regulatory C-terminal pseudo-kinase domain[6,27–30]. Phosphorylation of the MLKL pseudo-kinase domain by RIPK3 prompts an oligomerization event prior to plasma membrane translocation[28,31,32], where it enacts lytic permeabilization in a manner that is still currently debated[32–34].

Recent studies indicate there are important mechanistic differences between MLKL activation between species[25,27,29,35,36], although the extent of mechanistic diversity remains unclear considering MLKL orthologs are present in most chordate lineages[37,38]. The MLKL:RIPK3 cognate pair appear to have evolved as a signaling cassette[17], such that the recognition and phosphorylation of MLKL by RIPK3 is highly-speciated[35]. For example, despite >85% sequence identity between rat and mouse MLKL, rat MLKL is unable to reconstitute the necroptosis pathway in mouse cells lacking mouse MLKL[35]. Additionally, there appears to be important differences between the mechanisms underlying MLKL activation between species. In mouse cells, the executioner function of the 4HB domain is repressed by the MLKL pseudokinase domain until RIPK3-mediated phosphorylation relieves this interaction and triggers 4HB-directed cell death[28]. By contrast, in human cells, phosphorylation alone appears insufficient for MLKL activation[25,26]. Instead, human MLKL was proposed to additionally rely on a stable interaction with RIPK3 for its activation[17,25,26,29]. The nature of these interactions between MLKL and RIPK3, their chronology, and their role during necroptosis are incompletely understood and remain of immense interest.

We sought to dissect the nature and precise choreography of the human RIPK3:MLKL interaction during necroptosis signaling by developing synthetic protein ligands, termed Monobodies, that recognize distinct conformers of the MLKL pseudokinase domain. Monobodies are derived from a human fibronectin type-III domain scaffold, and bind to their target proteins with high affinity and high specificity[39,40]. The lack of disulfide bonds enables Monobodies to be expressed as genetically-encoded ligands in the reducing environment of the cytosol, where binding to their protein targets can be used to dissect signaling events[41,42]. We recently reported Monobodies that bind the human MLKL 4HB domain to inhibit necroptosis by

blocking recruitment of essential adapters for MLKL membrane translocation[31]. Here, we present Monobodies that recognize distinct human MLKL pseudokinase domain conformers, thus allowing the signaling events underlying RIPK3-mediated activation of MLKL to be mapped. We show that endogenous human MLKL and RIPK3 are pre-associated under basal conditions and, upon necroptotic stimulation and accompanying phosphorylation, MLKL disengages from RIPK3. Crystal structures of human MLKL pseudokinase domain in complex with Monobodies identified both open and closed forms in which the activation loop and αC helix adopts distinct conformations. Because monobody (Mb)27 recognition of MLKL is only observed following MLKL phosphorylation by, and release from, RIPK3, in cells, our structure identifies the closed form of MLKL containing an unstructured activation loop as the activated form of MLKL. Together, these studies reveal human MLKL dissociation from RIPK3 as an additional checkpoint in the necroptosis signaling pathway.

## Results

**Mb27 binding requires MLKL dissociation from RIPK3.** We developed monobodies that target the C-terminal pseudokinase domain of human MLKL: Mb(MLKL_26), Mb(MLKL_27), and Mb(MLKL_32) (referred to as Mb26, Mb27, and Mb32, respectively). Mb26 bound the human MLKL pseudokinase domain with a $K_D$ value of $176 \pm 13$ nM; Mb27 with $K_D$ value of $75 \pm 14$ nM (Supplementary Fig. 1a–c); Mb32 bound with a $K_D$ of $37.1 \pm 2.4$ nM, as previously reported[31]. To dissect protein interactions during necroptosis, we stably introduced monobodies under a doxycycline (dox)-inducible promoter into the human colon cancer cell line, HT29 (Supplementary Fig. 2a). These monobody constructs bear an N-terminal FLAG tag for detection of expression and immunoprecipitation, and a C-terminally fused GFP. Following dox-induced expression of these Monobodies, HT29 cells were exposed to a necroptosis stimulus (TSI: TNF (T); the Smac mimetic IAP antagonist, Compound A (S); and the pan-Caspase inhibitor, IDN-6556 (I)). In contrast to previously-reported monobodies directed toward the 4HB domain of human MLKL[31], Mb26, Mb27, and Mb32, which bind the pseudokinase domain, did not inhibit necroptosis (Supplementary Fig. 2b–d).

We next examined whether Mb26, Mb27, and Mb32 could bind MLKL within HT29 cells by immunoprecipitating these monobodies via their N-terminal FLAG tags and examining the pulled down proteins by immunoblot. Consistent with our earlier report[31], Mb32 bound human MLKL following treatment with the necroptotic stimulus, TSI, and additionally immunoprecipitated MLKL under basal conditions in the absence of stimulation (Fig. 1a). In contrast, Mb27 and Mb26 did not immunoprecipitate MLKL under basal conditions, and only recognized MLKL upon phosphorylation by RIPK3 following necroptotic stimulation with TSI (Fig. 1a; Supplementary Fig. 2e). We also examined whether MLKL was immunoprecipitated in complex with RIPK3. While Mb32-immunoprecipitated MLKL was bound to RIPK3 in both unstimulated cells and cells exposed to a necroptotic stimulus, Mb27-bound MLKL did not co-precipitate RIPK3 following treatment with the TSI necroptotic stimulus (Fig. 1a). Based on these data, we hypothesized that under basal conditions, MLKL is bound to RIPK3, both in the steady-state and following recruitment to the necrosome, and that RIPK3 binding to MLKL occludes the binding sites for Mb26 and Mb27. Consistent with this idea, we observed that Mb27 and Mb26 immunoprecipitated MLKL in the absence and presence of a necroptotic stimulus in $RIPK3^{-/-}$ HT29 cells (Fig. 1b; Supplementary Fig. 2f, g). Notably, Mb27 and Mb26 bound MLKL in $RIPK3^{-/-}$ HT29 cells even though MLKL was not phosphorylated (Fig. 1b), which was

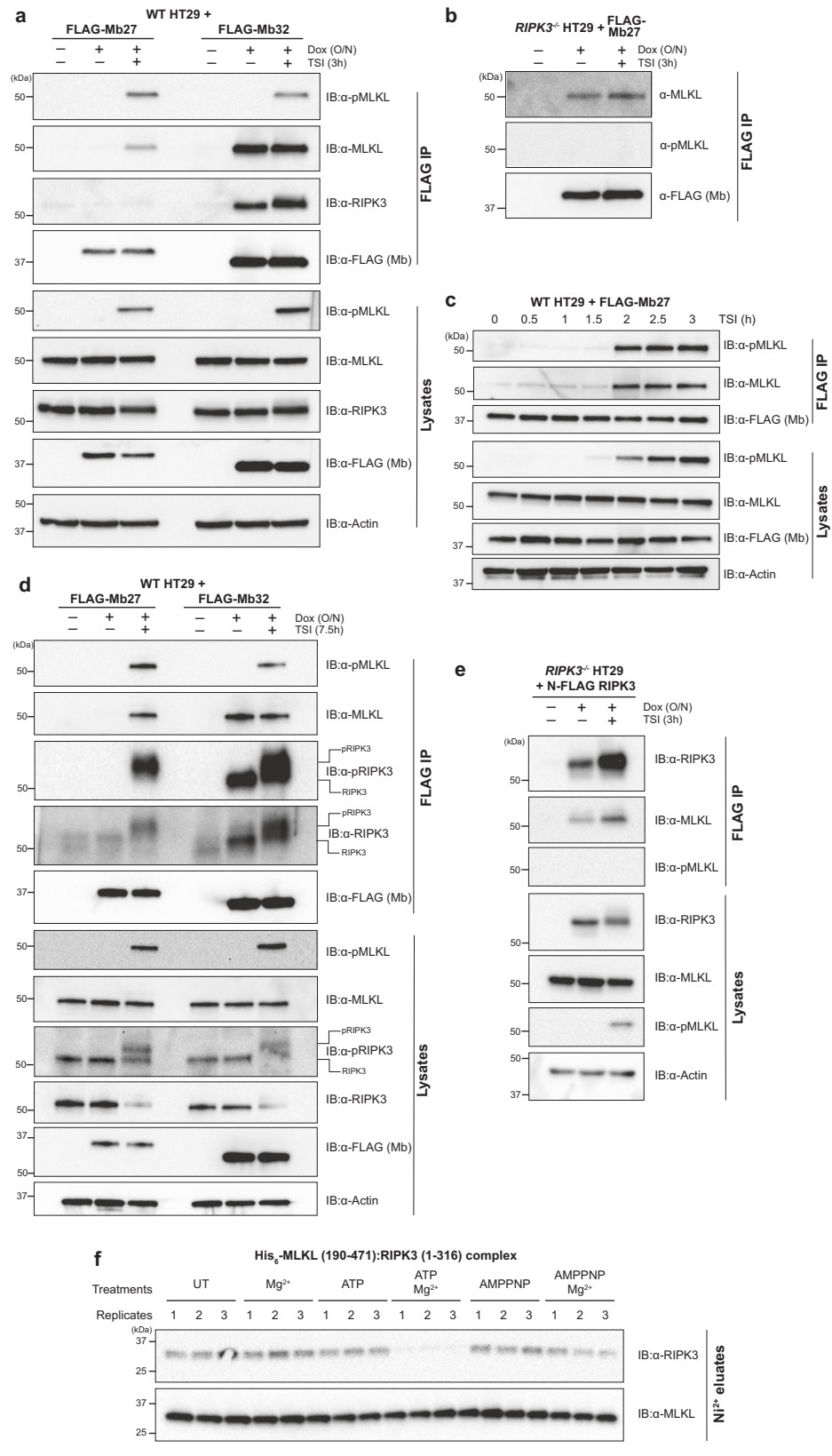

expected because RIPK3 is the only kinase identified thus far to phosphorylate the MLKL activation loop[43]. These data indicate that RIPK3 binding to MLKL under basal conditions occludes both Mb27 and Mb26 binding to MLKL, and that necroptotic stimuli induce MLKL disengagement from RIPK3 to present the Mb27 and Mb26 binding sites. We further investigated whether

Mb27 or Mb26 and RIPK3 compete for an overlapping binding site in MLKL using recombinant proteins. Here, we co-expressed and purified the heterocomplex of human RIPK3 kinase domain and the MLKL pseudokinase domain from insect cells and examined whether the purified MLKL:RIPK3 complex could be bound by His₆-tagged Mb26, Mb27, Mb32 and a

**Fig. 1 Mb27 binds activated human MLKL. a** Wild-type (WT) HT29 cells expressing Mb27 or Mb32 following doxycycline (Dox) induction were untreated (UT) or treated with the necroptotic stimulus, TSI (TNF; Smac mimetic, Compound A; and pan-Caspase inhibitor, IDN-6556), for 3 h and monobodies immunoprecipitated via their N-terminal FLAG-tag. Mb32 coimmunoprecipitated MLKL in both presence and absence of TSI stimulation, whilst Mb27 coimmunoprecipitated MLKL only in the presence of TSI. **b** Mb27 immunoprecipitated MLKL from lysates of $RIPK3^{-/-}$ HT29 cells in the absence and presence of necroptotic stimulation (TSI, 3 h). **c** Wild-type HT29 cells were treated with a necroptotic stimulus (TSI) over a 3 h timecourse, and Mb27 immunoprecipitation of MLKL examined. Mb27 co-immunoprecipitation of MLKL followed the emergence of RIPK3-mediated MLKL S358 phosphorylation (pMLKL) 1.5 h post-necroptotic stimulation. All images and blots are representative of at least two independent experiments. **d** FLAG-immunoprecipitates of Mb27 or Mb32 expressed (+Dox) in wild-type HT29 cells ±7.5 h TSI treatment revealed the presence of pRIPK3 bound to MLKL, with Mb27 immunoprecipitating MLKL only after TSI-stimulation. A representative of 4 independent experiments is shown. **e** FLAG-human RIPK3 expressed (+Dox) in $RIPK3^{-/-}$ HT29 cells coimmunoprecipitates MLKL in the presence or absence of TSI-stimulation, but not pMLKL post-TSI treatment. Lysates and IP samples were run on the same gel and subjected to the same exposure. A representative of independent duplicates is shown in each of (**c**, **e**). **f** Recombinant His$_6$-human MLKL pseudokinase domain and RIPK3 kinase domain complex expressed and purified from insect cells was untreated (UT) or incubated with combinations of $Mg^{2+}$, ATP or AMPPNP, the complex trapped by Ni-NTA chromatography and the eluates probed by immunoblot for RIPK3 and MLKL. Only ATP/$Mg^{2+}$ induced dissociation of RIPK3 and MLKL. Triplicates are shown for each condition.

previously-reported control monobody, Mb33, which recognizes the human MLKL 4HB domain[31] that is absent from MLKL in these constructs (Supplementary Fig. 3a–e). Strikingly, Mb26 and Mb27 could only bind the MLKL pseudokinase domain (Supplementary Fig. 3c, d), leading to dissociation of the MLKL: RIPK3 complex. In contrast, Mb32 bound MLKL alone and in complex with RIPK3 (Supplementary Fig. 3a, b), while, as expected, the control 4HB domain ligand, Mb33, did not recognize any form of the MLKL pseudokinase domain (Supplementary Fig. 3e). These data are concordant with our observations in human cells, and support the idea that RIPK3 engagement occludes the Mb27 and Mb26 binding site on the MLKL pseudokinase domain.

Recognition of MLKL by Mb27 mapped with the kinetics of RIPK3-mediated MLKL phosphorylation following treatment with the necroptotic stimulus, TSI. MLKL was bound by Mb27 two hours post-TSI treatment, following the emergence of MLKL phosphorylation 1.5 h post-TSI stimulation (Fig. 1c). Similarly, MLKL binding by Mb27 was observed at 7.5 h post-stimulation (Fig. 1d), which corresponds to the typical onset of HT29 cell death following treatment with the TSI necroptotic stimulus[32]. At 7.5 h post-TSI, we observed that phosphorylated RIPK3 (pRIPK3; upper band; antibody also recognizes unmodified RIPK3, which is the lower band[44]) was present in complexes immunoprecipitated by Mb27. We attribute this phenomenon to the assembly of MLKL oligomers at the necrosome into clusters prior to cell death, as we recently described[32,44], where phosphorylated MLKL (pMLKL) protomers are retained at the necrosome within an MLKL oligomer, but have disengaged from RIPK3 to expose the Mb27 binding site.

As these results indicate that phosphorylation of MLKL by RIPK3 leads to release of MLKL from the necrosome and exposure of the epitope for Mb27, we further examined the determinants of MLKL release. We reconstituted $RIPK3^{-/-}$ HT29 cells with FLAG-tagged human RIPK3 and immunoprecipitated RIPK3 complexes by FLAG pulldown. MLKL was observed in complex with RIPK3 under basal conditions, consistent with a previous overexpression study in HeLa and HT29 cells[4], as well as after 3 h of necroptotic stimulation (Fig. 1e). Notably, no phosphorylated MLKL was detected in complex with RIPK3, supporting the idea that phosphorylation destabilizes the MLKL:RIPK3 interaction. To further investigate this possibility, we examined the role of phosphorylation and binding of $Mg^{2+}$, ATP and the non-hydrolyzable ATP mimetic, adenylyl-imidodiphosphate (AMPPNP), on stability of the human RIPK3 kinase domain and His$_6$-tagged MLKL pseudokinase domain heterocomplex co-expressed and purified from insect cells (Fig. 1f). While ATP or $Mg^{2+}$ alone, or AMPPNP in the presence of $Mg^{2+}$, did not impact stability of the

heterocomplex, incubation of this complex with ATP and $Mg^{2+}$ led to the disengagement of MLKL and RIPK3. These data implicate the catalytic activity of RIPK3, which relies on the presence of both ATP and $Mg^{2+}$, rather than solely binding to ATP, AMPPNP or $Mg^{2+}$ alone, as driving the dissolution of the RIPK3 and MLKL heterocomplex. These findings provide compelling support for the idea that RIPK3-mediated MLKL phosphorylation serves as the cue for MLKL dissociation from the necrosome.

We further validated our observations from immunoblot studies that Mb32 binds MLKL constitutively, while Mb27 binds MLKL only post-necroptotic stimulation, by examining the Mb27 and Mb32 interactomes by mass spectrometry. Consistent with our immunoblot results, Mb32 bound MLKL and RIPK3 both under basal conditions and following treatment with the TSI necroptotic stimulus, whilst Mb27 displayed MLKL binding only after necroptotic stimulation and did not measurably bind RIPK3 (Supplementary Fig. 4a, b). Importantly, Mb32 immunoprecipitation of RIPK3 did not arise from non-specific binding; Mb32 did not capture RIPK3 from $MLKL^{-/-}$ HT29 cells (Supplementary Fig. 2h, i), confirming that Mb32-immunoprecipitated RIPK3 in complex with MLKL from wild-type HT29 cell lysates.

**Mb27 binding is centered on the human MLKL ATP-binding cleft.** To define the mechanism of action for Mb27, which binds human MLKL only after necroptotic stimulation, we crystallized the complex of Mb27 and the human MLKL pseudokinase domain (residues 190–471) and determined its crystal structure at 2.6 Å resolution (Fig. 2a–c, Table 1). This crystal structure revealed that Mb27 binds human MLKL via an epitope centered on the ATP-binding cleft of the pseudokinase domain (Fig. 2a, b). The human MLKL pseudokinase domain adopts a conventional, bilobal kinase fold, where the N-lobe comprises principally β-strands and the larger C-lobe mainly contains α-helices, with the ATP-binding cleft situated between these lobes. As expected, the MLKL pseudokinase domain structure closely resembles the previously determined high-resolution crystal structure of the apo human MLKL pseudokinase (Fig. 3a, b; PDB accession 4MWI, RMSD = 0.568 Å over 188 Cα atoms)[45]. The greatest deviation is in the N-lobe, which is shifted away from the C-lobe by 7.8° and opens a larger ATP-binding cleft between the lobes. Since Mb27 binds within this ATP-binding cleft, this modest structural change likely reflects a conformer stabilized by Mb27 binding. This does not change the overall topology, conformation, or preclude assembly of an intact regulatory (R)-spine (M254, I265, H329, F350) and the conserved salt-bridge between K230 and E250 that are characteristic of a closed, "active-like" kinase conformation (Fig. 3b; a representative electron density map is shown in Supplementary Fig. 5a).

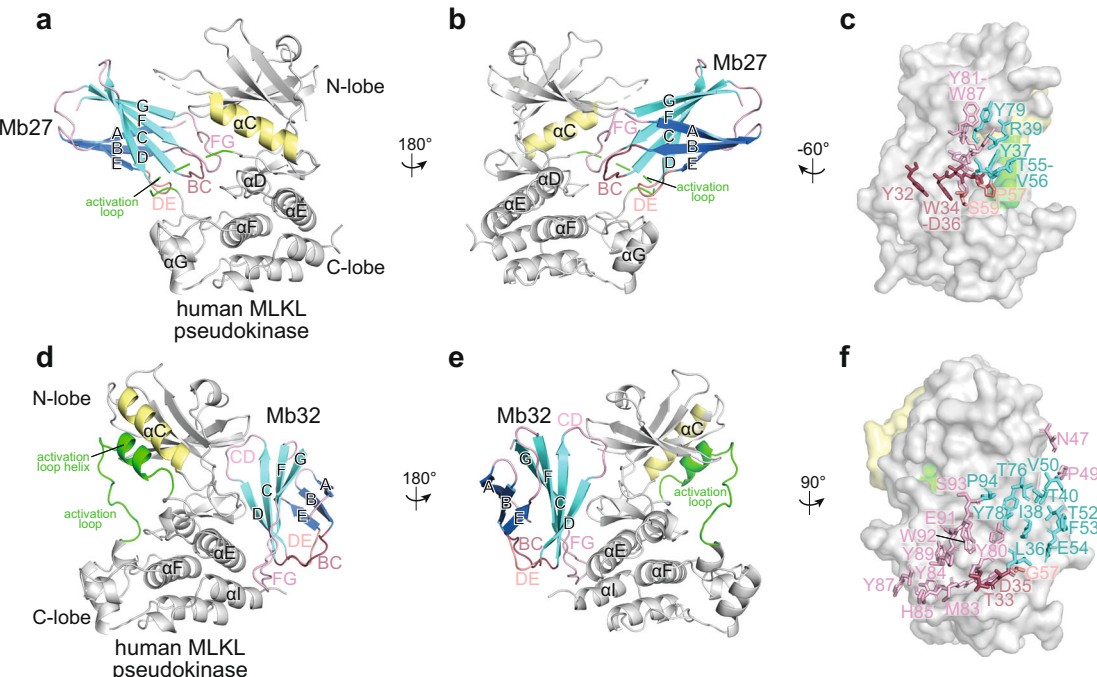

**Fig. 2 Mb27 and Mb32 bind distinct human MLKL epitopes. a, b, d, e** Transverse views of the monobody:human MLKL pseudokinase domain co-crystal structures. **a, b** Mb27 (β-sheets shown in light and dark blue) binds the ATP-binding cleft of the pseudokinase domain (gray; αC helix, yellow), atop the activation loop (residues 351-372; green), via its BC (raspberry), DE (salmon), and FG (light pink) loops. **c** Mb27 residues within 4.5 Å of MLKL are shown as sticks (colored as above). **d, e** Mb32 (β-sheets shown in marine and cyan) binds the hinge region of human MLKL pseudokinase domain (gray; αC helix, yellow; activation loop, green) principally via the Mb32 elongated FG loop, the CD loop and the D, G and F β-strands. **f** Mb32 residues within 4.5 Å of MLKL are shown as sticks (colored as above).

Mb27 primarily interacts with C-lobe residues, forms an interface area of 777 Å² and is estimated to release solvation free energy of 8.3 kcal/mol, as predicted by PDBePISA[46]. The interface area between other alternative pairs of monobody and MLKL in the crystal lattice is minimal, and a complexation significance score (CSS) of 1.0 at this interface is consistent with this being the biological assembly in solution. Mb27 originated from the "loop only" monobody library designed to form a convex binding surface[39] (Supplementary Fig. 1a), and accordingly primarily penetrates into the ATP-binding cleft of MLKL via its BC, DE, and FG loops (Fig. 2a, b), which fills in a large proportion of the space within the ATP-binding cleft. The primary interface is centered around the Mb27 BC loop, which binds the ATP-binding cleft side of the C-lobe, involving the αF-αG and β7-β8 loops, the C-terminus of the activation loop, as well as the αD helix. The Mb27 FG loop also forms a lesser interface with the loop preceding the β1 strand of the pseudokinase domain N-lobe. E213 and R333 of human MLKL appear to mediate the key electrostatic interactions by satisfying multiple hydrogen bonds and salt bridges with D36, R39, Y81, Y83, and W87 of Mb27 (Fig. 2c). At the centre of the C-lobe interface, W34 and Y83 of Mb27 interact with the sidechains of A375, Y376, I410, and P411 via hydrophobic interactions. To our knowledge, this is the first example of a monobody that binds to the respective active or pseudoactive site of a kinase or pseudokinase.

**Mb32 captures human MLKL in an open conformation.** To define the human MLKL binding mode of Mb32, which can pulldown MLKL and RIPK3 from cell lysates both under basal conditions and following post-necroptotic stimulation (Fig. 1a), we crystallized the human MLKL pseudokinase domain in complex with Mb32 and determined its crystal structure to 2.4 Å resolution (Fig. 2d–f, Table 1). Mb32 binds to an epitope centered

on the human MLKL C-lobe αE and αI helices and engages the β7–β8 loop at the hinge and the loop preceding the N-lobe β1 strand. This epitope is on the opposite face of the pseudokinase domain to the pseudoactive site to which Mb27 binds (Fig. 2a–c). Consistent with Mb32 originating from the "side and loop" monobody library designed to form a concave binding surface[39] (Supplementary Fig. 1a), the elongated FG loop, C- and D-strands, which are the principal sites of variation within this library, feature prominently in human MLKL binding (Fig. 2d–e). Accordingly, a series of FG loop residues (Y80, F82, M83, Y84, Y87, Y89) sit at the center of the epitope and bind V308, L311, Y318, L473, L466 of human MLKL via hydrophobic interactions (Fig. 2f). Mb32 binding buries a surface area of 1058 Å² and is estimated to release solvation free energy of 8.3 kcal/mol on average (between two copies of the interface in an asymmetric unit), as predicted by PDBePISA[46]. The interface area between other alternative pairs of monobody and MLKL in the crystal lattice is minimal, and a complexation significance score (CSS) of 1.0 at this interface supports this arrangement as the biological assembly in solution. The capacity of Mb32 to bind human MLKL both under basal conditions and following necroptotic stimulation indicates its epitope is accessible under both conditions. Whether Mb32 selectively recognizes the open conformer of MLKL is currently unclear, however. It is possible that Mb32 could bind via the hinge region of both the open and closed forms of MLKL, and that the conformational changes that occur during necroptotic signalling have minimal effect on Mb32 recognition of the hinge region. On the other hand, superimposing the open conformer of human MLKL (crystallized in complex with Mb32) on the human MLKL:Mb27 complex revealed extensive clashes, primarily with the activation loop, that would preclude Mb27 recognition of the open MLKL conformer (Supplementary Fig. 5b). Owing to these steric clashes, we, therefore, conclude

**Table 1 X-ray crystallography data collection and refinement statistics.**

| Structural parameters | Mb27:human MLKL (190–471) complex (PDB 7JW7) | Mb32:human MLKL (190–471) complex (PDB 7JXU) |
|---|---|---|
| Wavelength (Å) | 0.9537 | 0.9537 |
| Resolution range (Å) | 39.88-2.63 (2.724-2.63) | 36.56-2.441 (2.528-2.441) |
| Space group | P 3₁ 2 1 | P 4₃ |
| Unit cell (x, y, z, α, β, γ) | 64.805 64.805 226.763 90 90 120 | 94.619 94.619 115.197 90 90 90 |
| Total reflections | 34,476 (3379) | 74,917 (7164) |
| Unique reflections | 17,239 (1690) | 37,583 (3625) |
| Multiplicity | 2.0 (2.0) | 2.0 (2.0) |
| Completeness (%) | 99.19 (99.00) | 99.58 (96.69) |
| Mean I/sigma(I) | 11.96 (1.41) | 9.59 (1.00) |
| Wilson B-factor | 65.41 | 55.49 |
| R-merge | 0.02613 (0.3981) | 0.04141 (0.6141) |
| R-meas | 0.03696 (0.5629) | 0.05857 (0.8685) |
| R-pim | 0.02613 (0.3981) | 0.04141 (0.6141) |
| CC1/2 | 1 (0.929) | 0.995 (0.562) |
| CC* | 1 (0.981) | 0.999 (0.848) |
| Reflections used in refinement | 17,117 (1676) | 37,558 (3620) |
| Reflections used for R-free | 1717 (173) | 1992 (191) |
| R-work | 0.2460 (0.4402) | 0.2023 (0.3110) |
| R-free | 0.2684 (0.5230) | 0.2400 (0.3549) |
| CC(work) | 0.930 (0.817) | 0.959 (0.739) |
| CC(free) | 0.940 (0.599) | 0.962 (0.681) |
| Number of non-hydrogen atoms | 2782 | 6064 |
|   macromolecules | 2772 | 5966 |
|   ligands | | 16 |
|   solvent | 10 | 82 |
| Protein residues | 345 | 748 |
| RMS(bonds) | 0.012 | 0.003 |
| RMS(angles) | 1.75 | 0.55 |
| Ramachandran favored (%) | 97.63 | 99.59 |
| Ramachandran allowed (%) | 2.37 | 0.41 |
| Ramachandran outliers (%) | 0 | 0 |
| Rotamer outliers (%) | 24.84 | 2.55 |
| Clashscore | 3.61 | 3.66 |
| Average B-factor | 81.32 | 63.39 |
|   macromolecules | 81.36 | 63.52 |
|   MLKL | 81.68 | 59.58 |
|     N-lobe (190–285) | 73.93 | 66.64 |
|     C-lobe (286–471) | 86.03 | 55.92 |
|   Monobody | 80.44 | 75.81 |
|   ligands | | 61.64 |
|   solvent | 68.95 | 54.05 |

Statistics for the highest-resolution shell are shown in parentheses.

analogous closed, active-like conformation was also conserved in rat and horse MLKL pseudokinase domain structures[35] (Fig. 3d, e). Unexpectedly, the Mb32-bound human MLKL structure adopted an open conformation that resembles that of the mouse MLKL pseudokinase domain[6] (Fig. 3f). In this form, the R-spine and conserved β3 strand lysine:αC helix glutamate (K230:E250) salt-bridge are perturbed by the activation loop adopting a helical conformation, which displaces the αC helix. Accordingly, in the Mb32-bound structure, human MLKL adopts an open, inactive kinase-like conformation where the angle between N- and C-lobes is increased by 20.2° and the R-spine residues (M254, I265, H329, F350) are no longer aligned. Analogous to the mouse MLKL structure (PDB 4BTF[6]), where this conformation is stabilized by a hydrogen bond between K219 in the ATP-binding β3 strand and Q343 in the activation loop helix, a similar interaction is present within human MLKL between K230 and Q356 (Fig. 3c; representative electron density shown in Supplementary Fig. 5c). The similarity between the two MLKL structures strongly suggests that the human MLKL conformation captured in the Mb32:MLKL complex represents a low energy state within the conformational equilibrium of MLKL, rather than a high-energy or distorted state that is somehow induced by Mb32 binding. The low likelihood of capturing such high-energy structures with Monobodies or other crystallization chaperones has been detailed previously[49]. In mouse MLKL, disruption of the K219:Q343 bond triggers stimulus-independent MLKL activation leading to cell death by necroptosis[6]. By contrast, disruption of the analogous interaction in human MLKL upon introduction of the K230M[4] or Q356A (Supplementary Fig. 2j) did not trigger constitutive activation, consistent with the reported differences between mouse and human MLKL activation[25,27,29,35,36,45]. Similarly, introduction of the E250A substitution in human MLKL did not influence necroptotic signaling when expressed in $MLKL^{−/−}$ HT29 cells (Supplementary Fig. 2k). E250 resides in the αC helix and forms a salt-bridge with the β3 strand lysine, K230, that is synonymous with the closed, active-like conformation of protein kinases. These data indicate that additional interactions likely govern MLKL conformation, which can compensate for loss of Q356 or E250 in cellular contexts.

The Mb32-bound complex enabled us to model the complete structure of the human MLKL pseudokinase domain from residues 191 to 471, including the activation loop in which residues R353–S360 formed a short α-helix antiparallel to the αC helix within the N-lobe (electron density exemplified in Supplementary Fig. 5c). In previously-reported human MLKL structures[25,45,48], in addition to the Mb27-bound complex reported here, much of the activation loop, where the T357/S358 RIPK3 substrates reside, was not modeled due to lack of observed electron density. While the C-lobe of Mb32-bound MLKL is almost identical to that of the apo human MLKL pseudokinase domain (residues 286–471; RMSD = 0.296 Å, over 141 Cα atoms; Fig. 3g), the activation loop helix displaces the αC helix, resulting in a twist in the N-lobe by 20.2° that opens the ATP-binding cleft. The subsequent portion of the activation loop (residues L361–K372) runs adjacent to the pseudoactive site alongside the β1–β2 loop in the N-lobe and proximal to the αD helix in the C-lobe, which contributes D296 to a salt bridge with R370 in the activation loop.

Inspection of crystal packing did not implicate crystal contacts as the likely basis of this open conformation *in crystallo*. Molecular dynamics simulations indicated that within pMLKL, pS358 formed stable interactions with T246 on the αC helix (Fig. 3h), while pT357 was highly dynamic with a fully solvated side chain. Additionally, the activation loop helix is stable throughout these simulations and was not perturbed by phosphorylation of T357/S358 on the microsecond timescale

that Mb27 likely specifically recognizes the closed, active-like conformer, and not the open, activation loop helix form, of human MLKL.

Surprisingly, the pseudokinase domain in complex with Mb32 adopts a conformation distinct from the previously determined crystal structures of human MLKL pseudokinase alone (exemplified by PDB accession 4MWI[45], RMSD = 0.476 Å over 174 Cα atoms) or in complex with Mb27 (Fig. 3a–c, g). In all previous human MLKL structures[25,45,47,48], the pseudokinase adopts a characteristic closed kinase conformation, including an intact R-spine and a conserved salt bridge between K230 and E250. An

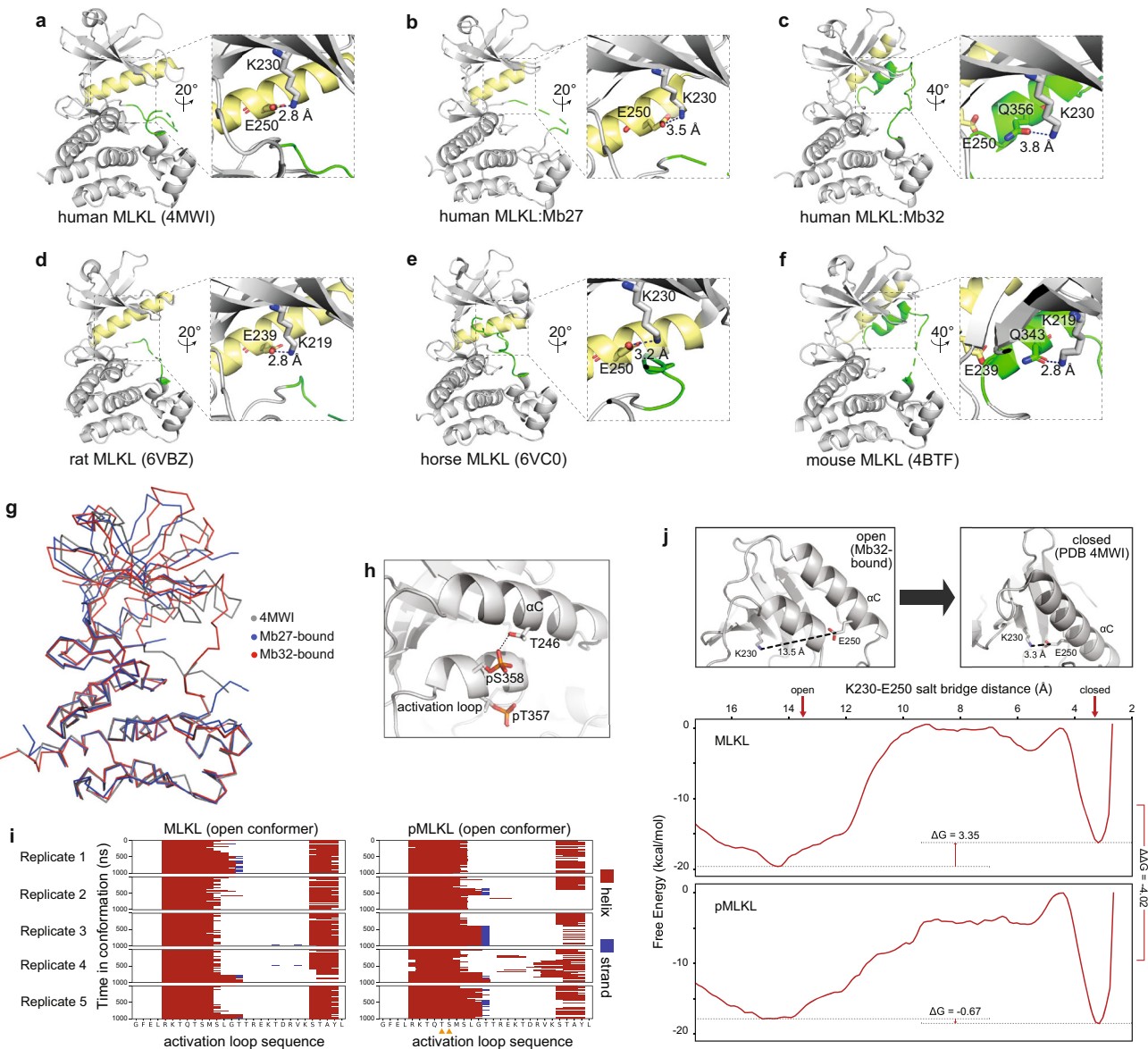

**Fig. 3 Human MLKL exists in two distinct conformations. a–f** A comparison of the unliganded human MLKL pseudokinase domain structure (**a**; PDB accession 4MWI)[45], the human MLKL pseudokinase domain from the Mb27 (**b**) and Mb32 (**c**) complexes and the rat (**d**; PDB accession 6VBZ)[35], horse (**e**; PDB accession 6VC0)[35]; and mouse (**f**; PDB accession 4BTF)[6] MLKL pseudokinase domain structures. Residues of the R-spine are shown as sticks in (**b**, **c**). Zoomed panels show the human MLKL K230–E250 (or equivalent residues in ortholog structures) salt bridge between the β3 strand ATP-binding lysine and the glutamate in the αC helix (yellow). A glutamine from the activation loop (green) helix hydrogen bonds with the β3 strand ATP-binding lysine in the open kinase conformation in mouse MLKL and the Mb32-bound human MLKL structure. **g** Overlaid Cα backbone traces of the previously-reported human MLKL pseudokinase domain structure (PDB 4MWI; gray)[45] and the Mb27- (blue) and Mb32 (red)-bound MLKL conformers reported herein. **h** Model illustrating the stable hydrogen bond between T246 on the αC helix and pS358 on the activation loop helix of human MLKL observed in molecular dynamics simulations. **i** Molecular dynamics simulations of the human MLKL pseudokinase domain open conformer containing the activation loop helix (from the Mb32 complex structure) at the microsecond timescale. Dephosphorylated MLKL is shown on the left and phosphorylated MLKL is shown on the right. Phosphorylated residues pT357 and pS358 are marked with oranges triangles below the sequence of pMLKL. Helices are marked in red and β-strands are marked in blue. Five independent replicates are shown. **j** The transition from open, Mb32-bound conformation to the closed conformation (from PDB, 4MWI[45]; top) was simulated and the two free energy surfaces along the reaction coordinate calculated as a function of the K230–E250 salt-bridge distance (lower). The ΔG was calculated as the difference between local free energy minima before and after 10 Å. Shown on the right, ΔΔG = −4.02 kcal/mol indicates the change in ΔG upon phosphorylation. A negative value suggests that phosphorylation stabilizes the closed, active-like conformation.

(Fig. 3i). Using enhanced sampling techniques, further MD simulations indicated that MLKL activation loop phosphorylation favors the transition of the MLKL pseudokinase domain from an open conformation with an activation loop helix to the closed form containing an unstructured activation loop (Fig. 3j). Free energy surfaces for MLKL and pMLKL were calculated as a function of the K230–E250 salt bridge distance using the

weighted histogram analysis method (WHAM)[50]. We used the K230–E250 salt bridge distance as a variable to quantify the transition from an open, inactive state (the Mb32-bound form) to a closed, active-like conformer (PDB ID: 4MWI)[45]. These results indicate that dephosphorylated MLKL favors the open Mb32-bound conformation (ΔG = 3.35 kcal/mol), while phosphorylated MLKL favors the closed conformation (ΔG = −0.67 kcal/mol).

Overall, the change in ΔG upon phosphorylation (ΔΔG = −4.02 kcal/mol) suggests that activation loop phosphorylation greatly increases the likelihood of MLKL adopting a closed, active-like conformation over the open conformation. It is important to note that, to date, only the phosphomimetic T357E/S358E MLKL, which exhibited the closed conformation[25], but no pMLKL, pseudokinase domain structure has been reported, and thus formal validation of these models awaits further experimentally-determined structures. Together, these data support the idea that the open conformation with a helical activation loop observed in the Mb32-bound human MLKL complex structure reflects an alternative state of human MLKL present in solution.

**Mb27 and RIPK3 bind overlapping sites on MLKL**. The underlying regulation of the interaction between MLKL and RIPK3 is of enormous interest. Because Mb27 selectively recognizes human MLKL only in the absence of RIPK3 or following necroptotic stimulation, Mb27 can be used as a tool to dissect regulation of the MLKL:RIPK3 interaction. To date, the only reported structure of an MLKL pseudokinase domain:RIPK3 kinase domain heterocomplex is that from mouse (PDB 4M69)[47], with no structures reported for the human MLKL:RIPK3 complex nor human RIPK3 kinase domain alone. The pseudokinase domains of mouse and human MLKL exhibit only 64.3% amino acid identity. The sequence of and around the activation loop harbors the most divergence, which includes the interface between mouse MLKL and RIPK3 C-lobes that was observed in the complex structure (Fig. 4a, b). We superimposed the human MLKL pseudokinase domain:Mb27 complex structure upon the mouse MLKL:RIPK3 complex (PDB accession 4M69; RMSD = 0.735 Å over 183 Cα atoms) to examine whether the Mb27 and RIPK3 binding epitopes overlap (Fig. 4c–d). This analysis revealed that the Mb27 and RIPK3 binding sites on human MLKL pseudokinase domain are located adjacent to one another, which partially overlap at the C-terminus of the activation loop (V371, K372, S373) and the N-terminus of the αG helix (S417). Therefore, we expect RIPK3 binding to MLKL will sterically-occlude Mb27 binding if the human MLKL:RIPK3 complex were analogous to the mouse MLKL:RIPK3 heterocomplex structure.

We identified 19 human MLKL residues within the Gly-rich loop and β2-β3 loop in the N-lobe, at the end of the catalytic and activation loops, on the αD helix and preceding the αG helix in the C-lobe as proximal to Mb27 in the crystal structure (Fig. 4c). To probe whether these residues might contribute to the RIPK3 binding site, we individually substituted each of these residues in full-length human MLKL with alanine and stably-introduced each mutant into $MLKL^{-/-}$ HT29 cells (Supplementary Fig. 6a). In parallel, we also introduced wild-type full-length human MLKL and the T357E/S358E mutant in which the RIPK3 substrate residues in the human MLKL activation loop are substituted with Glu to mimic phosphorylation. Following doxycycline treatment to induce expression, we examined the capacity of wild-type and mutant human MLKL constructs to reconstitute the necroptotic signaling pathway by quantifying SYTOX Green uptake as a measure of cell death using IncuCyte live cell imaging. Ala substitutions in the Gly-rich loop (R210A, E211A, N212A, and E213A), β2-β3 loop (Q236A, S239A), the catalytic (R333A, S334A, S335A) and activation (D369A, R370A, K372A, S373A, T374A) loops, within the αD helix (E293A, D296A) and preceding the αG helix (Q413A, N416A) did not measurably impact the necroptotic cell death after 20 h of necroptotic (TSI) stimulation (Fig. 4e). These mutants exhibited similar cell death kinetics to wild-type human MLKL (Supplementary Fig. 6b–u), indicating that, individually, each residue is dispensable for RIPK3 binding and activation. Because the death invoked by these mutants mirrored the activity of wild-type MLKL,

we conclude that these mutants can bind and be phosphorylated by RIPK3 to a comparable degree to wild-type MLKL. Intriguingly, only the αD helix mutant, R292A human MLKL, exhibited deficits in necroptotic signaling (Fig. 4e; Supplementary Fig. 6p). While these deficits are likely attributable to the disruption of key structural interactions (Fig. 4f), the extent of phosphorylated MLKL after necroptotic (TSI) stimulation was comparable that of wild-type MLKL (Fig. 4g). These data indicate that RIPK3 interaction was not perturbed, but instead downstream signaling by R292A MLKL is compromised. It is notable that R292A human MLKL was expressed at lower levels than most mutants, consistent with mutation compromising R292A stability by disrupting important structural interactions (Supplementary Fig. 6a). However, because the expression level was comparable to R370A human MLKL, which could reconstitute cell death signaling in $MLKL^{-/-}$ HT29 cells, deficits in R292A MLKL signaling are not likely attributable solely to a lower level of expression. Furthermore, R292A MLKL induced comparable phosphatidylserine (PS) exposure to wild-type MLKL and other mutants when Annexin V-binding was monitored by IncuCyte imaging. PS-exposure is known to precede necroptosis[51,52] and, because this was comparable between wild-type MLKL and all mutants, indicates MLKL expression is not likely to be a limiting factor (Supplementary Fig. 7a). This raises the possibility that although R292A MLKL can reach the plasma membrane, there may be deficits in assembly into the higher order hotspots responsible for membrane permeabilization and cell death. Consistent with previous studies[25,36], phosphomimetic Glu substitution of the MLKL activation loop residues, T357/S358, which are located adjacent to the Mb27 binding site (Fig. 4d), compromised reconstitution of the necroptotic pathway in $MLKL^{-/-}$ HT29 cells (Fig. 4e). While alanine substitution of T374 in human MLKL did not impact necroptotic signaling, introduction of the phospho-mimetic T374D mutation abrogated necroptotic signaling (Fig. 4e; Supplementary Fig. 6o) by impeding MLKL binding to, and phosphorylation, by RIPK3 (Fig. 4g). Human MLKL T374 was identified to be phosphorylated in a cell cycle-dependent manner[53,54], although at present the kinase responsible for this modification is unknown[43]. Our data indicate that T374, which resides within the Mb27-binding epitope on MLKL, contributes to RIPK3 engagement and necroptotic signaling, consistent with RIPK3 binding being centered on the adjacent MLKL activation loop residues, T357 and S358, and neighboring F386 in the αEF-αF loop[17,35] (Fig. 4d). To determine if the mutants, R292A and T374D, could still bind to Mb27, we stably expressed both Mb27 and MLKL mutants in $MLKL^{-/-}$ HT29 and performed Mb27 immunoprecipitations (Supplementary Fig. 7b). We detected no binding between Mb27 and R292A or T374D, validating the epitope identified in the Mb27:MLKL structure and further supporting the conclusion that the Mb27 and RIPK3 binding sites on MLKL partially overlap.

## Discussion

The implication of necroptotic cell death in the pathophysiology of a range of human diseases, including inflammatory diseases and those involving ischemic-reperfusion injury, has focused attention on therapeutically targeting the pathway. Because of its recent discovery, however, our current understanding of the pathway, including the molecular mechanism by which the terminal effector, MLKL, is activated by the upstream kinase, RIPK3, is incomplete. Here, we report the development of Monobodies that bind the human MLKL pseudokinase domain and serve as powerful tools for dissecting events in the chronology of necroptosis signaling. One monobody, Mb32, bound MLKL constitutively, and via Mb32 immunoprecipitation enabled us to validate the pre-association of endogenous MLKL and RIPK3 under basal conditions. Two further Monobodies, Mb26

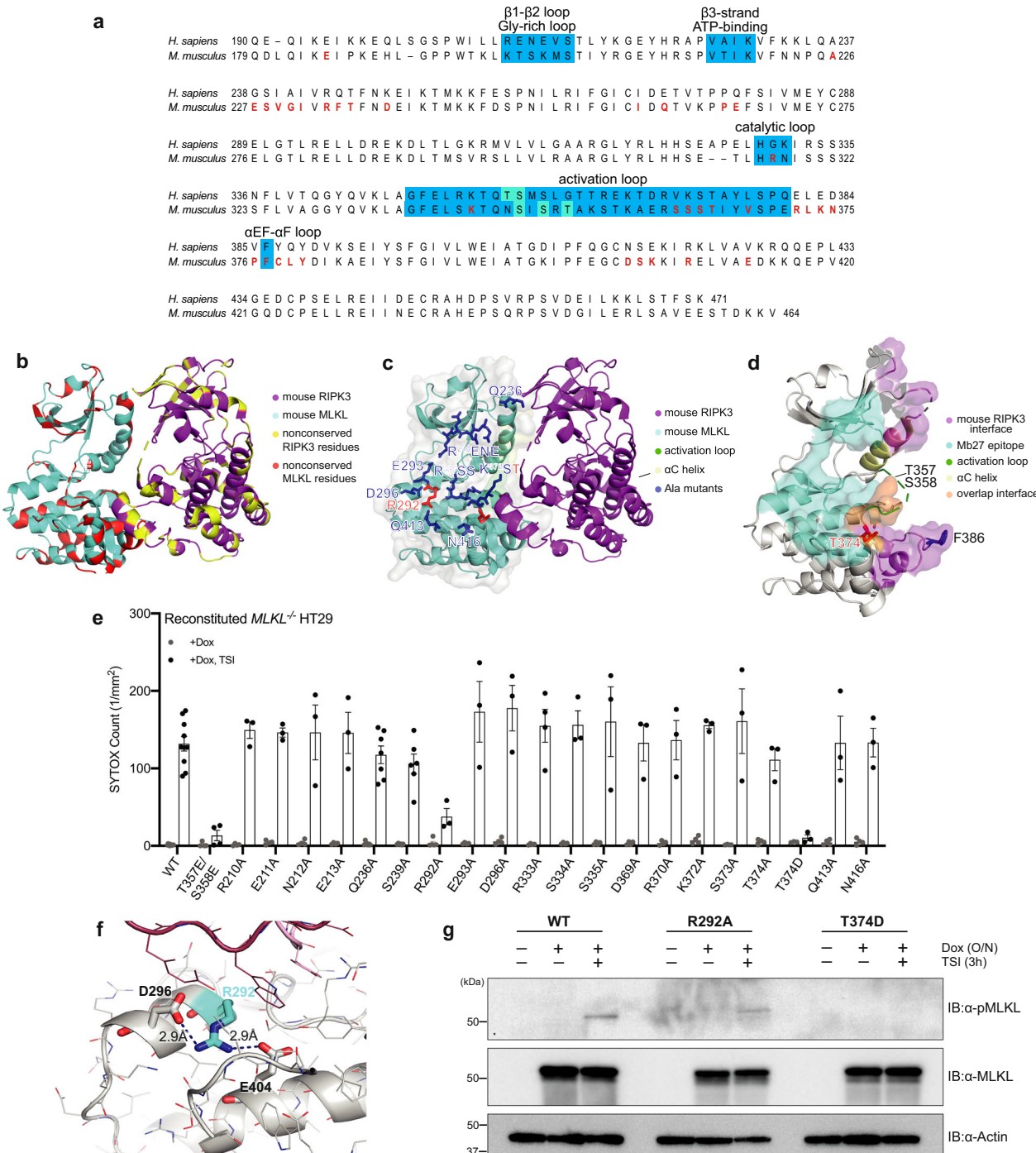

and Mb27, bound MLKL only after cells were exposed to a necroptotic stimulus, and their binding corresponded with MLKL phosphorylation and dissociation from the necrosome. As such, Mb26 and Mb27 can be used as molecular probes to monitor MLKL activation in human cells.

Our findings define two upstream events in the necroptosis signaling pathway that had been proposed, but lacked formal supporting evidence: (1) association of endogenous MLKL with RIPK3 under basal conditions; and (2) that MLKL phosphorylation leads to disengagement from RIPK3 following exposure to a necroptotic stimulus (Fig. 5). Here, we provide formal validation that endogenous RIPK3 and MLKL form a heterocomplex under basal conditions in human cells: an observation mirrored by the

stoichiometric complex formed by recombinant MLKL and RIPK3 proteins when recombinantly expressed and purified from insect cells. The existence of an MLKL:RIPK3 heterocomplex under basal conditions was inferred from previous studies in which over-expressed RIPK3 and MLKL coimmunoprecipitated from HeLa and HT29 lysates[4]; and from the observation that MLKL-like proteins encoded by some poxviruses bound the RIPK3 kinase domain and occluded cellular MLKL binding to disrupt necrop-totic signaling[17]. Recent studies indicate that after MLKL disen-gagement from necrosomal RIPK3, higher-order phosphorylated MLKL (pMLKL) complexes are trafficked to the plasma membrane[31] in vesicles via the Golgi-, actin- and microtubule-trafficking machinery[32]. The identities of the chaperones that

**Fig. 4 Mutation of MLKL R292 and T374 in the Mb27 binding epitope inhibits necroptosis. a** Sequence alignment of human and mouse MLKL pseudokinase domains. Residues of mouse MLKL that interact with RIPK3 in the MLKL:RIPK3 complex structure (PDB accession 4M69[47]; as predicted by PDBePISA[46]) in red text; key structural elements in teal; activation loop residues phosphorylated by RIPK3 in green. **b** The mouse MLKL pseudokinase domain (cyan):RIPK3 kinase domain (purple) co-crystal structure (PDB accession 4M69)[47] with non-conserved residues in human MLKL and RIPK3 in red and yellow, respectively. **c, d** The human MLKL:Mb27 structure was superimposed on the mouse MLKL:RIPK3 complex structure (PDB 4M69), with Mb27-proximal residues identified for characterization in **e** shown as sticks (**c**). **d** The Mb27-binding interface on the human MLKL pseudokinase (cyan surface), residues within 4.5 Å of the superimposed mouse RIPK3 kinase domain (PDB 4M69) (purple surface) and overlapped residues in the Mb27- and mouse RIPK3-epitopes, V371, K372, S373, and S417 (orange surface) are shown. The αC helix (yellow), and previously implicated RIPK3-interacting regions of human MLKL, the activation loop (green) and F386 (blue sticks)[17,35], and T374 implicated herein, are highlighted. **e** Evaluation of necroptotic signaling by wild-type and Mb27 binding epitope mutants of full-length human MLKL in MLKL−/− HT29 cells. Wild-type or mutant human MLKL expression was induced with doxycycline (Dox) and cell death was measured by SYTOX Green uptake (1/mm²) quantified using IncuCyte S3 live cell imaging in the presence or absence of a necroptotic stimulus (TNF, Smac mimetic, IDN-6556; TSI) for 20 h. Two independent cell lines were generated for WT, S239A, Q236A, and one for other MLKL mutants; WT lines were assayed in n = 5 independent experiments. TSEE, R333A and one independent Q236A line were assayed in n = 4 independent experiments; other mutants in n = 3 independent experiments. Data are plotted as mean ± SEM. **f** Alanine substitution of MLKL R292 likely disrupts key interactions with neighboring residues. **g** Wild-type and R292A, but not T374D, human MLKL expressed in MLKL−/− HT29 cells were phosphorylated in response to 3 h TSI treatment. Data are representative of duplicate independent experiments.

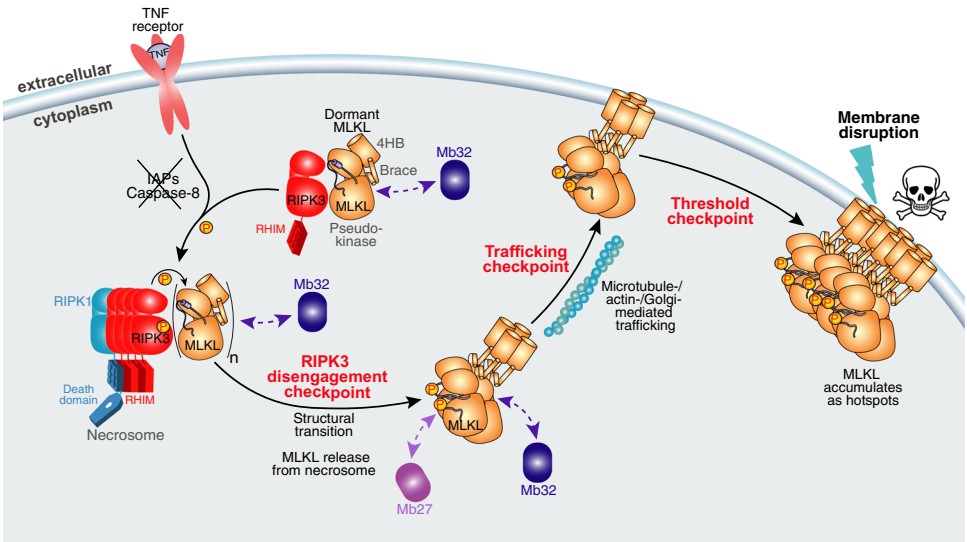

**Fig. 5 Summary of regulatory checkpoints in human MLKL activation.** Necroptosis is initiated by ligand binding to cell surface death receptors, such as TNF receptor 1, when the IAP E3 ubiquitin ligases and Caspase-8 are depleted or inhibited. Immunoprecipitation using Mb32 identified MLKL to exist in complex with RIPK3 prior to initiation of necroptosis. The RIPK3:MLKL complex is recruited to a RIPK1-nucleated platform (the necrosome) following induction of necroptosis, where RIPK3 is autophosphorylated and MLKL phosphorylated by RIPK3. Subsequently, oligomeric phosphorylated MLKL (pMLKL) disengages from RIPK3, which permits its recognition by Mb27. Phospho-MLKL oligomers are subsequently trafficked to the plasma membrane[31] via the Golgi-/actin-/microtubule-trafficking machinery[32]. pMLKL accumulates at the plasma membrane into higher order hotspots, and when a threshold amount of MLKL is surpassed, MLKL ruptures the plasma membrane to induce cell death and release of pro-inflammatory DAMPs[32]. Monobody-27 and Monobody-32 are depicted in light and dark purple colors, respectively; dashed lines represent the interactions observed for Mb27 and Mb32 in HT29 cells. The skull and crossbones image (Mycomorphbox_Deadly.png; artist, Sven Manguard) was sourced under a Creative Commons Attribution-Share Alike 4.0 license.

traffic MLKL to the plasma membrane are yet to be discovered, although we pinpointed their site of interaction as the α4 helix of the N-terminal executioner four-helix bundle domain of MLKL using the inhibitory monobodies, Mb33 and Mb37, in our previous study[31]. Once pMLKL reaches the plasma membrane, a critical threshold amount of MLKL must be reached via assembly of pMLKL membrane "hotspots" to achieve membrane disruption and consequent lytic cell death[32]. Several proteins and post-translational modifications have been reported to modulate MLKL activity to date (reviewed in ref. [29,43]), and it is likely that further auxiliary proteins contribute to tuning of MLKL's killer function[34,55]. With the availability of monobodies that can detect pMLKL release from the necrosome (Mb27 and Mb26, reported here) and block pMLKL membrane translocation (Mb33, Mb37), we expect future studies will be enabled to dissect which regulated steps in the necroptosis pathway, which

we term checkpoints, are influenced by modulatory proteins and post-translational modifications.

By solving the structures of human MLKL pseudokinase domain in complex with Mb27 and Mb32, we identified two distinct MLKL conformers. Mb32 enabled crystallization of a human MLKL conformer in which the activation loop formed an atypical helix that buttresses against, and displaces, the αC helix, leading to an open conformation. This conformation is reminiscent of the mouse MLKL pseudokinase domain[6], which to date has only been observed in this form. Now, with the illustration that human MLKL can adopt a similar conformation, additional support is offered to the ideas that MLKL behaves as a biphasic switch[45] and, more broadly, that pseudokinases can function as dynamic modulators in signal transduction[56–58]. Molecular dynamics simulations indicate this conformation to be stable on the microsecond timescale, while enhanced sampling methods

indicate phosphorylation serves to lower the energy barrier for the transition of MLKL from the open form with an activation loop helix to the closed, active-like conformation. Accordingly, we reasoned that the open conformation of human MLKL pseudokinase domain likely reflects the inactive, RIPK3 binding conformation for several reasons. Firstly, mouse MLKL adopts a similar conformation in its complex with the mouse RIPK3 kinase domain[47], raising the possibility that human MLKL interacts with human RIPK3 via an analogous mode. Secondly, Mb27, which binds the pro-necroptotic form of MLKL in cells, bound the closed MLKL conformer in which the site of RIPK3-mediated phosphorylation—the activation loop – was unstructured. Notably, Mb27 bound human MLKL via an epitope that overlaps the activation loop and, owing to the mutual exclusivity of Mb27 and RIPK3 binding, allowed us to pinpoint a role for the MLKL activation loop in RIPK3 engagement. In addition, previous studies have highlighted the activation loop as a dynamic element in MLKL vertebrate orthologs, with phosphorylation proposed to trigger conformational interconversion to a pro-necroptotic form and/or necrosomal RIPK3 dissociation[35]. Our structural studies indicate that Mb27 is a specific probe for detecting the closed, active-like conformer of MLKL, which emerges following necroptotic stimulation, RIPK3-mediated phosphorylation of MLKL, and MLKL disengagement from the necrosome. The precise molecular basis underlying RIPK3 recognition of MLKL and how RIPK3 maintains MLKL in an inactive form awaits high resolution structural analyses.

This study is a second example of monobody-assisted studies of (pseudo)kinases in which monobodies developed under the same experimental conditions captured two distinct conformations of (pseudo)kinases. In a previous study of Aurora A kinase, Monobodies targeted to its allosteric effector-binding site captured active and inhibited forms of the kinase domain[59]. In solution, these conformations of Aurora A kinase are populated at substantial levels in the absence of a bound monobody[60]. The fact that Mb27 and Mb32 were developed from the same monobody screen under identical conditions strongly suggests that these two conformations represent major conformers of MLKL in solution and points to the broad utility of monobodies in identifying and capturing distinct conformers of an inherently dynamic proteins, such as signaling kinases[49,61].

Collectively, our findings offer formal support for the ideas that RIPK3 binds and maintains human MLKL in an inactive form under basal conditions, and that necroptotic stimulation triggers MLKL phosphorylation, conformational interconversion, and dissociation from RIPK3. By mapping the early events in MLKL activation that precede MLKL membrane translocation and plasma membrane disruption, our studies have unveiled additional signaling checkpoints that could be targeted pharmacologically to inhibit pathological necroptotic cell death.

## Methods

**Generation of monobodies.** For monobody screening, bait protein was generated from a bacmid prepared in DH10MultiBac *E. coli* (ATG Biosynthetics) from a pFastBac Htb (Thermo) vector encoding a TEV (tobacco etch virus) protease-cleavable His$_6$ fusion N-terminal to the human MLKL pseudokinase domain (residues 190–471), which was C-terminally fused to a non-cleavable flexible penta-Ser linker and AviTag (ASSSSSGLNDIFEAQKIEWHE) using the Bac-to-Bac system in Sf21 insect cells, as reported for untagged counterparts[25,36,45]. The AviTag of human MLKL pseudokinase domain was biotinylated using recombinant His-BirA using established protocols[31], and biotinylated protein eluted from a Superdex S200 10/300 GL size-exclusion chromatography column in 0.2 M NaCl, 20 mM HEPES pH 7.5, 5% v/v glycerol. The bait was immobilized by binding to Streptavidin MagneSphere Paramagnetic Particles (Promega) and subjected to four rounds of screening of monobody phage-displayed libraries, before enriched sequences were subcloned into a yeast display library and subjected to two further rounds of yeast displayed screening, using established procedures[31,39,62].

**Expression constructs.** Monobody sequences were amplified by PCR from library clone templates and ligated into a derivative of the doxycycline-inducible, puromycin-selectable vector, pF TRE3G PGK puro, encoding an N-terminal FLAG and C-terminal GFP sequence (synthesized by ATUM, CA). For immunoprecipitation of mutant MLKL proteins, monobody sequences were introduced into the doxycycline-inducible, hygromycin-selectable vector, pF TRE3G PGK hygro, encoding an N-terminal FLAG and C-terminal GFP sequence. Mutations were introduced into a human MLKL DNA template (from DNA2.0, CA) using oligonucleotide-directed overlap PCR or were synthesized by ATUM (CA) and subcloned into pF TRE3G PGK puro as BamHI-EcoRI fragments. A gene encoding N-terminally FLAG-tagged human RIPK3 was synthesized by ATUM (CA) and ligated into pF TRE3G PGK puro. Insert sequences were verified by Sanger sequencing (Micromon DNA Sequencing Facility, VIC, Australia). Oligonucleotide sequences are listed in Supplementary Table 1. Vector DNA was co-transfected into HEK293T cells with pVSVg and pCMV ΔR8.2 helper plasmids to generate lentiviral particles, which were transduced into HT29 or U937 cells and selected for genomic integration using puromycin (2.5 μg/mL; StemCell Technologies) using established procedures[6,25,28].

**Reagents and antibodies.** Primary antibodies used in this study were: rat anti-MLKL (clone 3H1, produced in-house; 1:1000 dilution; available as MABC604, EMD Millipore, Billerica, MA, USA), rat anti-human MLKL pseudokinase domain (clone 7G2, produced in-house; 1:2000 dilution)[32], rabbit anti-human MLKL phospho-S358 (AB187091, Abcam; 1:3000), mouse anti-Actin (A-1978, Sigma-Aldrich, St Louis, MO, USA; 1:5000), rat anti-human RIPK3 (clone 1H2, produced in-house; 1:1000)[17], rabbit anti-human phospho-S227 (D6W2T, CST, 1:1000), mouse anti-FLAG M2 (Sigma-Aldrich, F1804, 1:3000). Recombinant hTNF-Fc, produced in-house, and the Smac mimetic, Compound A, have been previously described[63,64]. The pan-caspase inhibitor, IDN-6556/emricasan, was provided by Tetralogic Pharmaceuticals.

**Generation of *RIPK3*$^{−/−}$ HT29 cells.** HT29 cells expressing humanised *S. pyogenes* cas9[25] were transduced with a lentivirus encoding a doxycycline-inducible sgRNA sequence specifically targeting the second exon of RIPK3 (GAATTCGTGCTGCGCCTAGA) via a lentiviral vector, pCMV ΔR8.2, in which the viral integrase gene was mutated by oligonucleotide-directed PCR to introduce the D64V mutation to ablate integrase activity. This vector was packaged with a pVSVg helper plasmid in HEK293T cells to generate lentiviral particles. *RIPK3* editing was initiated using an established procedure[25]; sgRNA expression was induced using doxycycline (1 μg/mL), cells were sorted as pools by flow cytometry for high expression of sgRNA vector-encoded GFP after two days, and after a further week, cells were single-cell sorted based on absence of GFP to confirm absence of integration. Effective knockout of RIPK3 was confirmed by culturing in TSI to ensure non-responsiveness to necroptotic stimuli, using anti-RIPK3 western blot and next-generation sequencing (Illumina).

*Cell culture.* The human colorectal adenocarcinoma HT29 (a gift from Mark Hampton; and their *MLKL*$^{−/−}$ [25] and *RIPK3*$^{−/−}$ [this work] counterparts) were cultured in human DMEM (Gibco) media and human monocytic lymphoma *MLKL*$^{−/−}$ U937 cells were cultured in human RPMI media. DMEM and RPMI media were supplemented with 8% vol/vol fetal calf serum (FCS; Sigma), with puromycin (2.5 μg/mL; StemCell Technologies) added for lines stably transduced with inducible MLKL or monobody constructs. Routine PCR testing confirmed cell lines to be mycoplasma-negative.

**Flow cytometry cell death assay.** HT29 cells were plated $3.75 \times 10^4$ cells/well in 24-well plate, allowed to settle 16–24 h and then treated with doxycycline (20 ng/mL) for 3 h to induce expression of the relevant monobody construct prior to stimulation. U937 cells were plated at $5 \times 10^4$ cells/well into 96-well plates and treated with doxycycline, as above, overnight to induce exogenous protein expression. *MLKL*$^{−/−}$ U937 and HT29 cells were then treated with doxycycline (20 ng/mL) in combination with TNF (100 ng/mL), the Smac-mimetic compound A (500 nM) and the pan-caspase inhibitor IDN-6556 (5 μM) to induce necroptosis. Cell death was quantified by propidium iodide (PI; 1 μg/mL; Sigma #P4170) uptake using flow cytometry 24 h post-stimulation. Data were combined from 3 to 5 replicate experiments and presented as mean ± SEM. Data were collected using a BD FACSCalibur instrument and analysis was performed using FlowJo (v10.1r7) using the gating strategy depicted in Supplementary Fig. 2l, m.

**IncuCyte cell death assay.** HT29 cells were seeded into 48-well plates at $3 \times 10^4$ cells/well and left to settle for 16–24 h prior to treatment with doxycycline (20 ng/mL) overnight or for 3 h to induce expression of the relevant MLKL construct. Cells were then treated with TNF (100 ng/mL), the Smac-mimetic compound A (500 nM) and the pan-caspase inhibitor IDN-6556 (5 μM) to induce necroptosis in phenol red-free media supplemented with 1% FCS, 1 mM Na pyruvate (ThermoFisher Scientific #11360070), 1 mM L-GlutaMAX and SYTOX Green (ThermoFisher Scientific #S7020) or AlexaFluor488-conjugated Annexin V (ThermoFisher Scientific #A13201) as before[32]. Cells were moved into the IncuCyte S3 System (Essen Bioscience) and imaged using default bright-field and green

channel settings on 10x objective. Scans were obtained every hour for 20 or 24 h, and number of SYTOX Green- or Annexin V- positive cells per mm$^2$ over time was quantified using IncuCyte S3 v2018A software (Essen Bioscience).

**Western Blot**. HT29 cells were seeded into 24-well plates at $7 \times 10^4$ cells/well and induced overnight with 20 ng/mL doxycycline for Mb or MLKL expression, or 2.5 ng/mL doxycycline for RIPK3 expression. Cells were harvested in 2× SDS Laemmli lysis buffer, sonicated, boiled at 100 °C for 10–15 min, and then resolved by 4–15% Tris-Glycine gel (Bio-Rad). Proteins were transferred to nitrocellulose or PVDF membrane and probed with antibodies as indicated.

**Immunoprecipitation for western blot**. HT29 cells seeded into 10 cm plates at $2.5 \times 10^6$ cells/plate or 6-well plates at $3.5 \times 10^5$ cells/well were induced overnight with 20 ng/mL or 1 μg/mL doxycycline and treated with TNF (100 ng/mL), Smac-mimetic (Compound A; 500 nM) and pan- caspase inhibitor IDN-6556 (5 μM) (TSI) for 3 or 7.5 h. Cells were harvested in lysis buffer (50 mM Tris-HCl pH 7.4, 1% (v/v) Triton X-100, 1 mM PMSF, 1 mM EDTA, 150 mM NaCl, 2 mM Sodium Vanadate, 10 mM NaF, Complete protease inhibitor tablet (Roche)) and super-natants were mixed with Anti-FLAG M2 Affinity Gel (Millipore). Beads were washed in lysis buffer and proteins harvested using 2× SDS lysis buffer or eluted with 1 mg/mL FLAG peptide. FLAG elution samples were mixed with 5X SDS lysis buffer. Samples were boiled at 100 °C for 5 min, then resolved by 4–15% Tris-glycine gel (Biorad). After transfer to nitrocellulose or PVDF, membranes were probed with antibodies as indicated.

**Mass spectrometry**. HT29 cells seeded into 15 cm plates at $10^7$ cells/plate were induced overnight with 20 ng/mL doxycycline and treated with TNF (100 ng/mL), Smac-mimetic (Compound A; 500 nM) and pan-caspase inhibitor IDN-6556 (5 μM) (TSI) for 3 h. 3 × 15 cm plates per condition were harvested in lysis buffer (50 mM Tris-HCl pH 7.4, 1 % (v/v) Triton X-100, 150 mM NaCl, 1 mM EDTA, 2 mM sodium vanadate, 10 mM NaF, 1 mM PMSF, Complete protease inhibitor tablet (Roche)) and supernatants were mixed with Anti-FLAG M2 Affinity Gel (Millipore). Beads were washed in lysis buffer and proteins eluted with FLAG peptide (2×0.5 mg/ml elutions and 1× 1 mg/mL elution). Quadruplicate samples were resuspended in 6 M Urea, 10 mM TCEP and 100 mM Tris-HCl pH 7.0 and digested using filter aided sample preparation, lyophilized and analyzed on a M-Class UPLC (Waters, USA) coupled to a timsTOF Pro (Bruker) mass spectrometer with a CaptiveSpray source. Peptides resuspended in 2% ACN, 1% formic acid (FA) were separated on a 25 cm × 75 μm analytical column, 1.6 μm C18 beads with a packed emitter tip (IonOpticks, Australia) at 40 °C, 400 nL/min using a gradient from buffer A (99.9% MilliQ water, 0.1% FA) to buffer B (99.9% ACN, 0.1% FA; 55 min) as previously[31]. Data collected on the timsTOF Pro (Bruker) were searched against the human Uniprot Reference Proteome with isoforms (downloaded March 2019). Inclusion relied on proteins being identified in at least 50% of the replicates in one group; probability values were corrected for multiple testing using Benjamini–Hochberg method. Analyses were performed using Limma (v3.40.6) in R (v3.6.1).

**Recombinant protein expression and purification**. Human MLKL pseudokinase domain (residues 190–471) was expressed and purified using established procedures[45] in Sf21 insect cells using the Bac-to-Bac system (Invitrogen). Briefly, human MLKL (190–471) was cloned into pFastBac Htb to include an N-terminal, TEV-cleavable His$_6$ tag, and a bacmid was prepared from DH10MultiBac E. coli (ATG Biosynthetics). The human MLKL pseudokinase domain (residues 190–471): human RIPK3 kinase domain (residues 1–316; harboring C3S and C111A sub-stitutions to prevent disulfide bond formation) complex was co-expressed in Sf21 insect cells and purified following the same protocol. The co-expression bacmid was prepared in DH10MultiBac E. coli (ATG Biosynthetics) from a pFastBac Htb vector encoding a TEV protease-cleavable His$_6$ fusion N-terminal to the human MLKL pseudokinase domain (residues 190–471), with an additional fragment containing p10 promoter, the gene encoding human RIPK3 (residues 1–316; C3S, C111A), and HSV TK poly(A) signal (derived from pAceBac2; ATG Biosynthetics) introduced into pFastBac Htb:human MLKL (190–471) by In-Fusion (Takara Bio) cloning into the non-coding region between BbsI and SnaBI sites of the pFastBac Htb vector backbone. 0.5 L Sf21 cells at a density of $2.5 \times 10^6$ cells/mL cultured in Insect-XPRESS media (Lonza) in 2.8 L Fernbach flasks shaking at 27 °C, 90 rpm were infected with 10% v/v P2 baculovirus and cultured for a further 48 h. Cells were pelleted by centrifugation at $500 \times g$ and snap frozen in LN$_2$. Thawed pellets were resuspended in wash buffer (0.5 M NaCl, 20 mM Tris-HCl pH 8, 20% v/v glycerol, 10 mM imidazole pH 8) supplemented with 0.5 mM TCEP (Pierce) and EDTA-free Complete Protease inhibitor (Roche), and lysed by sonication. Debris was eliminated by centrifugation and supernatant applied to HisTag Ni-NTA resin (Roche) with gentle agitation for 1 h at 4 °C before beads were pelleted and washed with wash buffer and wash buffer containing 35 mM imidazole. Proteins were eluted in a buffer equivalent to wash buffer, but containing 250 mM imidazole pH 8. Following TEV protease cleavage of the His$_6$ tag, the protein was dialyzed into 0.2 M NaCl, 20 mM HEPES pH 7.5, 5% v/v glycerol before adding to fresh HisTag resin. The unbound material was concentrated by centrifugal spin ultrafiltration and applied to a HiLoad 16/600 Superdex-200 pg size-exclusion column (Cytiva).

Fractions containing the protein of interest were identified by reducing SDS-PAGE with stain-free imaging (Biorad), pooled, spin concentrated, and frozen in liquid N$_2$ for storage at −80 °C if not used fresh. Monobodies with in-frame TEV (tobacco etch virus) protease cleavable N-terminal His$_6$ tags (in pProEX Htb; Invitrogen) were expressed in E. coli BL21-Codon Plus (DE3)-RIL and crude lysate was extracted using established methods[31]. The clarified lysate in 0.5 M NaCl, 20 mM Tris-HCl (pH 8), 20% v/v glycerol, 10 mM imidazole (pH 8), 0.5 mM TCEP [tris-(2-carboxyethyl)phosphine] was loaded onto a HisTrap HP 5 ml column (Cytiva) pre-equilibrated with low imidazole buffer [0.5 M NaCl, 20 mM Tris-HCl (pH 8), 20% v/v glycerol, 5 mM imidazole (pH 8)] at 4 °C. After washing in low imidazole buffer, the bound protein was eluted over a linear gradient over 30 mL into high imidazole buffer [0.5 M NaCl, 20 mM Tris-HCl (pH 8), 20% v/v glycerol, 250 mM imidazole (pH 8)]. The eluted protein was further purified by cleaving the His$_6$ tag by incubating with TEV protease, dialysis, Ni-chromatography to eliminate uncut material and the TEV protease, and snap frozen for storage at −80 °C.

**HisTrap protein interaction assay**. To probe the ability of each monobody to bind to purified recombinant MLKL:RIPK3 complex, 0.2 mg purified His-tagged monobody was mixed with 0.2 mg untagged recombinant MLKL:RIPK3 in low imidazole buffer [0.5 M NaCl, 20 mM Tris-HCl (pH 8), 20% v/v glycerol, 5 mM imidazole (pH 8)] and loaded onto a HisTrap HP column (Cytiva) pre-equilibrated in the same buffer. After washing in low imidazole buffer, the bound protein was eluted over a linear gradient over 30 mL into 100% high imidazole buffer [0.5 M NaCl, 20 mM Tris-HCl (pH 8), 20% v/v glycerol, 300 mM imidazole (pH 8)] and analyzed by reducing SDS-PAGE using Stain-free imaging (Biorad, CA). To detect the dissociation of recombinant MLKL:RIPK3 complex in response to ATP/AMPPNP/Mg$^{2+}$ treatments, 20 μg His-tagged MLKL (190–471):untagged RIPK3 (2–316) was mixed in 40 μL low imidazole buffer supplemented with 10 mM MgCl$_2$, 1 mM ATP or 1 mM AMPPNP at 4 °C for 1 h The pre-treated proteins were loaded onto 40 μL Ni-NTA slurry (HisTag resin, Roche) pre-equilibrated in the corresponding low imidazole buffers and incubated at 4 °C for 1 h with gentle agitation. After three washes in 100 μL of corresponding low imidazole buffers, the bound protein was eluted into 80 μL high imidazole buffer containing the relevant additives. Because His-tagged MLKL pseudokinase domain migrates equivalently to the RIPK3 kinase domain by SDS-PAGE, elutions were diluted 500-fold and analyzed by immunoblot using rat anti-human RIPK3 kinase domain (clone 1H2; 1:1000 dilution) and rat anti-human MLKL pseudokinase domain (clone 7G2; 1:2000 dilution). Assays were performed in triplicate.

**Protein crystallisation and structure determination**. Recombinant human MLKL pseudokinase domain (190–471) and either Mb27 or Mb32 were coeluted by Superdex 75 10/300 GL size-exclusion chromatography (Cytiva) and con-centrated to 5 mg/ml using a 10 kDa MWCO spin concentrator. Crystallization trials were conducted at the C3 Facility (Parkville, VIC) with 150 nL protein solution mixed with 150 nL reservoir volume in sitting drops, with 50 μL reservoir volume. Mb27:MLKL pseudokinase domain crystals grown in 0.2 M magnesium chloride, 20% w/v PEG 3350 at 8 °C, and Mb32:MLKL pseudokinase domain crystals grown in 0.1 M Bis-Tris chloride (pH 6.5), 20% w/v PEG monomethyl ether 5000 at 8 °C were soaked in cryoprotectant (reservoir solution supplemented with 25% v/v ethylene glycol) and flash-cooled in liquid nitrogen. X-ray diffraction data were collected at the Australian Synchrotron MX2 beamline using the EIGER X 16M detector[65] using the inhouse MX beamline control software. Diffraction data were indexed and integrated in XDS[66], and merged and scaled in AIMLESS[67]. Phases were solved by molecular replacement using the published human MLKL pseudokinase domain structure (PDB accession 4M67)[47], Mb33 structure[31] (PDB accession 6UX8, for Mb27), and another monobody structure[68] (PDB accession 5N7E, for Mb32) as search models in PHASER[69]. COOT and phenix.refine were used for iterative real-space and reciprocal-space refinement by manual model building and phase refinement[70,71]. After initial molecular replacement, part of the structural model deviated significantly from the observed electron density in the N-lobe of MLKL in complex with Mb32, as it adopts a different conformation to the search model. This part of the model was deleted and manually rebuilt in COOT through iterative real-space and reciprocal-space refinement by manual model building and phase refinement[70,71]. Protein interfaces and the biological assembly in solution were analyzed using PDBePISA[46]. Interface area is defined as "the difference in total accessible surface areas of isolated and interfacing structures divided by two" in Å$^2$. Structure models were validated using MolProbity[72]. All structural figures were generated using PyMOL (The PyMOL Molecular Graphics System, Schrödinger, LLC.).

**Molecular dynamics**. The MLKL pseudokinase domain coordinates from the Mb32 complex were used as the starting conformation for molecular dynamics simulations. Post-translational modifications were introduced using PyTMs[73]. Unbiased all-atom molecular dynamics (MD) simulations were performed using GROMACS 2016.4[74]. Structures were parameterized using the CHARMM36[75] force field and solvated with the TIP3P water model. Random solvents molecules were replaced with sodium or chloride ions to neutralize the charge of the system. The system was contained in a dodecahedron at least 1 nm larger than the protein from all sides with periodic boundary conditions. Long-range interactions were

calculated with particle mesh Ewald. Neighbor lists were maintained using the Verlet cutoff scheme[76]. The system underwent steepest descent minimization until the maximum force was <100 kJ/mol. Canonical ensemble[77] was used to heat the system from 0 to 310 K in 100 ps. Isothermal-isobaric ensemble[78] (1 bar, 310 K) was applied for 100 ps. Positional restraints were applied during equilibration. Production runs used 2 fs time steps. Enhanced sampling methods were implemented using PLUMED-patched GROMACS[79]. Structures of dephosphorylated and phosphorylated MLKL were minimized and equilibrated as previously described for unbiased simulations. Steered MD simulations were used to estimate intermediate conformations between the human MLKL open, inactive (MLKL-Mb32) and closed, active state (PDB ID: 4MWI)[45]. Moving restraints for steering included a combination of dihedral, center-of-mass distance, and RMSD restraints which were manually optimized to accurately steer the system to the active state. All moving restraints were implemented as a harmonic potential with a center moving linearly over time which were added to the system Hamiltonian. Umbrella samples were used to calculate free energy surfaces with the Weighted Histogram Analysis Method (WHAM) as implemented by WHAM v2.0.10 (Alan Grossfield). Free energy surfaces were calculated as a function of the K230–E250 salt bridge distance which was defined as the distance between two atoms: K230-NZ and E250-CD. We simulated umbrella samples from 110 intermediate states each dephosphorylated and phosphorylated MLKL spanning a range of K230:E250 salt bridge distances from ~3 to 18 Å. Umbrella sampling was implemented as a harmonic potential with a static center defined by the initial KE salt bridge distance and a spring constant parameter of 60. These potentials were added to the Hamiltonian of each umbrella sample. Umbrella samples were simulated for 80 ns each for a total of (110 × 80 ns) of simulation time for each dephosphorylated and phosphorylated MLKL.

**Reporting summary**. Further information on research design is available in the Nature Research Reporting Summary linked to this article.

## Data availability

Data, including expression construct sequences, are available from the corresponding authors upon request. The atomic coordinates for the human MLKL pseudokinase domain complexes with Mb27 and Mb32 have been deposited in the Protein Data Bank with the accession numbers, 7JW7 and 7JXU, respectively. Source data are provided with this paper.

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

## Acknowledgements

We thank staff at the Australian Synchrotron MX beamline for assistance with data collection. This research was undertaken in part using the MX2 crystallography beamline at the Australian Synchrotron, Victoria, Australia, and made use of the ACRF Detector. We acknowledge scholarship support for S.E.G. and A.V.J (Australian Government Research Training Program Stipend Scholarships), S.E.G. (Wendy Dowsett Scholarship), Y.M. (Melbourne Research Scholarship and AINSE PGRA scholarship). We are grateful to the National Health and Medical Research Council for fellowship (J.M.H., 1142669; P.E.C., 1079700; J.M.M., 1105754, 1172929), grant (1057905; 1124735, 2002965) and infrastructure (IRIISS 9000653) support; Australian Cancer Research Foundation; and the Victorian Government Operational Infrastructure Support scheme. Funding for N.K. from NIH (5R01GM114409 and 1R35GM139656) is acknowledged.

## Author contributions

S.E.G., Y.M., and A.K. designed and performed experiments, and analyzed data; J.J.S. performed mass spectrometry experiments and analyzed data with the assistance of A.I.W.; E.D., A.V.J., A.L.S., C.R.H., C.F., S.N.Y., P.P.C.S., E.J.P., J.M.H., and J.M.M. performed experiments and analyzed data; W.Y. and N.K. performed and analyzed molecular dynamics simulations; J.M.H., P.E.C., S.K., and J.M.M. supervised the project and contributed to experimental design and data analysis. S.E.G., Y.M., S.K., and J.M.M. co-wrote the paper with input from all authors.

## Competing interests

S.E.G., A.L.S., C.F., S.N.Y., E.J.P., J.M.H., P.E.C., and J.M.M. contribute, or have contributed, to a project developing necroptosis inhibitors in collaboration with Anaxis Pharma Pty Ltd. A.K. and S.K. are listed as inventors on issued and pending patents on the Monobody technology filed by The University of Chicago (US Patent 9512199 B2 and related pending applications). The other authors declare no competing interests.
