## [Peer Review File · Nature Communications]

REVIEWER COMMENTS

Reviewer #1 (Remarks to the Author):

In this manuscript, Garnish et. al. screened out two monobodies which specifically bind human MLKL pseudo-kinase domain. Using these monobodies, they got two crystal structures of MLKL(pseudo-kinase domain)-monobody complexes. Interestingly, the structures of MLKL in these complexes showed differences: the monobody-27(Mb27) binding MLKL showed a closed, “active-like” kinase conformation with a conserved salt-bridge between K230 and E250. It is similar as previous human MLKL structure (PDB:4MWI) and rat (PDB:6VDZ) or horse (PDB:6VC0) MLKL structure. While the monobody-32(Mb32) binding MLKL showed an open conformation with a unique salt-bridge between K230 and Q356, which is similar as mouse MLKL structure. Based on these crystal structures and some biochemical experiments using monobodies as tools, the author claimed conformational interconversion of MLKL and disengagement from RIPK3 was required and could be a check point for the following events of MLKL activation. Although the model is interesting, but the evidences in this manuscript are not strong enough to support this model.

1. By co-precipitation assay, the author found Mb27 did not bind MLKL when RIP3 was present but not activated (Figure 1a, lane-2). While when RIP3 was knock-out, Mb27 could bind MLKL well (Figure 1b). But pervious literature (such as Sun et al, Cell, 2012) showed that before necroptosis is induced, RIP3-MLKL binding is week which is because the RIP3 kinase activity and auto-phosphorylation of site S227(which is required for MLKL binding) should be elevated by necroptosis induction. It means that most MLKL is free and not in complex with RIP3 before necroptosis is induced. So, it is confused that why Mb27 did not bind these free MLKL. I notice that the expression level of Mb27 is lower than Mb32 in wild-type HT-29 cells (showed in Figure 1a and Figure S1a), while expression level of MB27 in RIP3^{-/-} HT-29 cells is comparable with Mb32 in wild-type HT-29 (showed in Figure 1b and Figure S1d). The authors should use a new stable HT-29 cell-line with higher expression level of Mb27 with similar expression of Mb32 to test the possibility that detection of the free MLKL in Mb27-precipitates before necroptosis-induction is due to or not due to Mb27 expression level.

2. In Figure 1a and S3, the author analyzed the Mb27-MLKL binding at the time-point 3 hours after necroptosis induction. The Figure S4 showed that the necroptosis began at 4 hours after induction in HT-29 cells. And the previous paper from same group submitting this manuscript have showed that in HT-29 cells the phosphorylation level of MLKL at 6-hour after necroptosis induction was about 5 to 20 folds higher to phosphorylation level of MLKL at 3-hour after induction (Figure 1C from Tanzer, M.C. et al. Biochem J , 2015). Therefore, 3-hour is not enough for these assay, time-course assay containing 6-hour and later time-points is required for Figure 1 and S3. And NSA is recommended if the author wants to test the longer time-point such as 12-hour or 20-hour later from necroptosis induction.

3. The author claimed disengagement of phosphor-MLKL (pMLKL) from RIPK3 can be a checkpoint for MLKL activation and cell necroptosis (in Figure 5). The checkpoint usually means a very important event in biological process, such as the checkpoints in cell cycle. It indicated disengagement of pMLKL from RIPK3 is necessary for MLKL function. But current data have just shown there was free phosphor-MLKL in cytosol and not in complex with RIP3. It may come from the unphosphorylated MLKL competition of the binding of RIP3 to release the pMLKL. And it may come from the PPM1b removing the pS227 of RIP3 in the complex to decrease RIP3-MLKL binding affinity to release the pMLKL. The author should

give more evidences to support the model in Figure 5 that disengagement of pMLKL from RIPK3 is a checkpoint step for MLKL activation and preceding necroptosis. Or it just plays an important (or a small, or no) role in preceding necroptosis.

4. In figure S1b, the author showed the K_d of Mb27 binding to MLKL. I recommend the author may test K_d of RIP3-MLKL binding in vitro. If the K_d results show RIP3-MLKL binding is stronger than Mb27-MLKL, it may explain why Mb27 do not bind MLKL when RIP3 is present before necroptosis induction.

5. MLKL conformation is different in Mb27-MLKL and Mb32-MLKL crystal structure. But the author mentioned that Mb32 may bind “both the open and closed forms of MLKL” (in line 185 of the manuscript). Is it possible that Mb27 also bind both the open and closed forms of MLKL? If Mb27 only binds to closed forms of MLKL. It should figure out why in RIP3-KO cells, there is enough closed forms of MLKL to bind Mb27, while with RIP3 expression the closed forms of free MLKL are gone. Did it mean most of MLKL are in complex of RIP3 in untreated cells? The author is recommended to use enough beads containing RIP3 antibody to pull-down RIP3-MLKL complex, and see how much MLKL is in complex with RIP3 and how much MLKL is left being free in cytosol before necroptosis induction. And the free MLKL in cytosol and can be used to test Mb27-binding.

6. The title “Conformational interconversion of MLKL and disengagement from RIPK3 precedes cell death by necroptosis” seems to indicate the conformational interconversion contribute to MLKL disengagement from RIPK3. If the answer is yes, It is interesting to figure out how open to close form transition contributes MLKL disengagement from RIP3. Q356A mutant disrupts an important salt-bridge in open form of MLKL. So that, it is supposed to have more closed form of MLKL than wild-type of MLKL. It is recommended to test the binding affinity (K_d) in vitro of wild-type and Q356A with RIP3 and Mb27. If Mb27 only binds with closed form of MLKL, and open to close form transition will weaken the RIP3-MLKL binding and induce disengagement of MLKL from RIP3. It may get a result that the binding to RIP3: WT > Q356A ; and the binding to Mb27: WT < Q356A. Otherwise, conformational interconversion of MLKL and disengagement from RIPK3 may be unrelated events.

7. E250-K230 salt-bridge is conserved in human, rat horse MLKL and Mb27-MLKL closed form structure. It is interesting to test if E250 mutation has effect on necroptosis proceeding, RIP3-binding and Mb27 binding.

8. In figure 4 and S4, the author showed alanine mutation in the site R292 in the Mb27 binding epitope prevent necroptosis. Did it disrupt or enhance the binding to Mb27? How about other mutations in Mb27 binding epitope? Mb27 expression did not affect necroptosis proceeding (Figure S2b). So, how R292A prevent necroptosis? Did R292A function to prevent open to closed form transition and disengagement from RIP3? The current cell death results seem not contribute to the model of MLKL in Figure 5. The author should give more related data of the R292A and other mutants for MLKL conformational transition and disengagement from RIP3. Otherwise, these cell death results of MLKL mutants were only confusing but not contributed to the model in Figure 5.

Reviewer #2 (Remarks to the Author):

Murphy et al. have come up with interesting facts in their study aimed at investigating the conformational transformations observed in pseudokinase domain of MLKL after phosphorylation by RIPK3. This is an interesting study as post-phosphorylation, MLKL undergoes oligomerization to induce cellular necroptosis. The authors have used two monodies to investigate the conformational transformations observed by MLKL after phosphorylation by RIPK3. For that purpose they developed monobodies that specifically target pseudokinase domain of MLKL Mb27 and Mb32 (with KD values of 75 nM and 37.1 nM respectively) The authors have successfully shown that whereas MB32 complexed with MLKL pseudokinase domain, as illustrated by their immunoprecipitation assays, the MB27 immunoprecipitated only with phosphorylated MLKL by RIPK3. These assays showed that necroptotic stimuli induce structural transformations which lead to disengagement of MLKL from RIPK3 consequently presenting the Mb27 binding site. Therefore phosphorylation of MLKL by RIPK3 and its disengagement is pivotal for its role in necroptosis. To investigate the conformational transformations exhibited by MLKL pseudokinase domain, the authors also co-crystallized MB27 and MB32 with MLKL pseudokinase.

The authors have observed two distinct open and closed MLKL conformations in these complexes which are mediated by the movement of MLKL activation loop across alpha-c helix. The authors also successfully showed that whereas Mb32 complex structure exhibits open conformation, MB27 binds with pro-necroptotic closed form of MLKL.

With aforementioned experiments and analyses, the authors have successfully addressed their questions aiming to investigate structural and functional transformations exhibited by MLKL pseudokinase and role of RIPK3 in inducing these transformation in cellular necroptosis.

The article is well written and experiments have been clearly presented and conclusions have been logically drawn, therefore I recommend the article for publication.

Reviewer #3 (Remarks to the Author):

The manuscript entitled "Conformational interconversion of MLKL and disengagement from RIPK3 precedes cell death by necroptosis" is generally well-written. The authors do a good job of highlighting the novelty and importance of their findings, as well as supporting their conclusions with clearly presented data. The study presents the first experimental evidence that human MLKL, the terminal effector in necroptotic signaling, behaves as a conformationally regulated switch protein. The conclusions rely heavily upon the observed differential binding behaviors of two distinct Monobodies to MLKL, and the structural differences discerned in two separate co-crystal structures.

The two most important advantages of this study are: 1) the demonstration of different conformations of human MLKL, whereas previously only structures from other species demonstrated this difference and 2) data from cells which support that different conformations of MLKL are represented in different signaling states.

The two crystal structures are of sufficiently high quality for publication, but inclusion of a few additional statistics and figures are requested.

A drawback of the study in its current form is how binding of Mb27 occludes RIPK3 binding to MLKL. The authors provide structure-based mapping that suggest the binding sites overlap somewhat, yet more evidence would make a stronger argument. For example, direct

evidence of competition between Mb27 and RIPK3 would support an overlapping binding site. Alternatively, assessment of the R292A mutant MLKL's ability to bind to RIPK3 and Mb27 would add value to the conclusion. Also, while the other alanine mutant MLKL constructs did not show changes in cell death kinetics, whether they had nevertheless demonstrated diminished binding to RIPK3 and/or Mb27 was not assessed. It seems possible that binding to these mutants could be reduced but the downstream effect on cell death is not noticeably changed. While performing these experiments may be outside the scope of the current study, at the very least, a more thorough discussion of the overlap between the presumed RIPK3 and Mb27 binding sites would benefit the strength of the argument. The following points are also related:

- How similar are the human and mouse MLKL sequences at the RIPK3 binding site? When discussing how the human MLKL/RIPK3 interface was modelled using the mouse structure, it would be informative to include the %identity/similarity between mouse and human at that interface.
- What is the rmsd upon superposition of human MLKL from Mb27-bound structure with mouse MLKL from RIPK3-bound structure?

Other points and questions:

- In RIPK3 knockout cells, Mb27 is able to bind MLKL, suggesting that MLKL can adopt the closed/active conformation in the absence of phosphorylation. Or alternatively, does another kinase phosphorylate MLKL to achieve this closed conformation in cells lacking RIPK3? Or does MLKL exist in this closed conformation in the basal state (in equilibrium with the open conformation)? Have the authors probed the Mb27 IP/MLKL coIP in RIPK3 knockout cells for phospho-MLKL?
- Lines 140 and 172, please remove "supplementary"
- The comparisons of the different conformations of the pseudokinase domain across the structures would benefit from a simple superposition of the structures from Figures 3a-b-c as a simple C-alpha trace, preferably by superposing the C-lobes in order to better show the differences in N-lobe orientation with respect to the C-lobe.
- Please include B factor information in Table 1 for monobodies and MLKL separately. In addition, it would also be of value to include the B factors for the N- and C-lobes of the pseudokinase domains separately.
- Could the authors consider including a representative region of electron density, either 2Fo-Fc and/or an omit map, for particular regions of interest in each structure? For example: the area including activation loop/helix and the hydrogen bond between K230 and Q356 from the Mb32 structure, and between K230 and E250 in the Mb27 structure; alternatively, the electron density at the interface of the pseudokinase domain and Monobodies would demonstrate the quality of the maps.
- Does the crystal packing reveal any other kinase/Monobody pair that could represent another interface that might exist in solution?
- The Ramachandran distribution is a bit problematic, with less than 98% in the favored region in both structures. At these modest resolutions, the geometry could be improved some through tighter restraints during refinement, though would likely not change the overall conclusions of the study.
- The authors should briefly elaborate in their Methods section on the search models used for molecular replacement. Was the molecular replacement search model for MLKL a single kinase domain, or was it split into N- and C-lobes and searched individually? At these modest resolutions, and with large conformational differences between structures, a clearer picture of how the solutions were obtained could be very helpful for a reader.

We thank the reviewers for their constructive comments and positivity towards our study. We have revised our manuscript extensively to accommodate the reviewers' suggestions (changes highlighted in main text with grey shading). Additionally, we have added Figure 1d-f and additional data in panel b, Figure 3g and j, Figure 4a, b, f, g and additional data in panel e; Suppl. Fig. 1 was revised and a panel added, Suppl. Fig. 2a was modified, and panels d-g and k added, Suppl. Fig. 4 and Suppl. Fig. 5 were added, Suppl. Fig. 6o added and panel a modified, Suppl. Fig. 7a was modified and panel b added. We address each reviewer query (*blue italics*) with point-by-point responses (black, plain text) below.

REVIEWER COMMENTS

Reviewer #1 (Remarks to the Author):

In this manuscript, Garnish et. al. screened out two monobodies which specifically bind human MLKL pseudo-kinase domain. Using these monobodies, they got two crystal structures of MLKL(pseudo-kinase domain)-monobody complexes. Interestingly, the structures of MLKL in these complexes showed differences: the monobody-27(Mb27) binding MLKL showed a closed, "active-like" kinase conformation with a conserved salt-bridge between K230 and E250. It is similar as previous human MLKL structure (PDB:4MWI) and rat (PDB:6VDZ) or horse (PDB:6VC0) MLKL structure. While the monobody-32(Mb32) binding MLKL showed an open conformation with a unique salt-bridge between K230 and Q356, which is similar as mouse MLKL structure. Based on these crystal structures and some biochemical experiments using monobodies as tools, the author claimed conformational interconversion of MLKL and disengagement from RIPK3 was required and could be a check point for the following events of MLKL activation. Although the model is interesting, but the evidences in this manuscript are not strong enough to support this model.

1. By co-precipitation assay, the author found Mb27 did not bind MLKL when RIP3 was present but not activated (Figure 1a, lane-2). While when RIP3 was knock-out, Mb27 could bind MLKL well (Figure 1b). But pervious literature (such as Sun et al, Cell, 2012) showed that before necroptosis is induced, RIP3-MLKL binding is weak which is because the RIP3 kinase activity and auto-phosphorylation of site S227(which is required for MLKL binding) should be elevated by necroptosis induction. It means that most MLKL is free and not in complex with RIP3 before necroptosis is induced. So, it is confused that why Mb27 did not bind these free MLKL.

We thank the reviewer for prompting further discussion of this important point. We feel our data prompt a rethink of the paradigm for how MLKL is maintained in a dormant form in the cytosol, because it is clear from studying endogenous proteins that MLKL is complexed with RIPK3 before activation. It is important to note that the earlier study of Sun used overexpressed RIPK3 and MLKL to examine their interaction, which will not faithfully reflect the endogenous stoichiometry. Using Mb32, we observed MLKL in complex with RIPK3 both by immunoblot and by mass spectrometry. Considering that MLKL is bound to RIPK3 under basal conditions, and that RIPK3 binding overlaps the site of Mb27 binding on MLKL, we would not expect Mb27 to bind unless RIPK3 is removed from the system and MLKL can toggle to the closed, active-like form.

Based on the reviewer's query, we have emphasised this view throughout the revised manuscript and have presented additional supporting data examining Monobody interactions

with recombinant MLKL:RIPK3 complexes (Supp. Fig. 3), which mirror our findings in HT29 cells as discussed on page 14, lines 387-389.

I notice that the expression level of Mb27 is lower than Mb32 in wild-type HT-29 cells (showed in Figure 1a and Figure S1a), while expression level of MB27 in RIP3^{-/-} HT-29 cells is comparable with Mb32 in wild-type HT-29 (showed in Figure 1b and Figure S1d). The authors should use a new stable HT-29 cell-line with higher expression level of Mb27 with similar expression of Mb32 to test the possibility that detection of the free MLKL in Mb27-precipitates before necroptosis-induction is due to or not due to Mb27 expression level.

We thank the reviewer for the opportunity to present further supporting data that illustrate expression level is not the basis for our observations. Below we present IP results from a different construct of Mb27 which contains a serendipitous frameshift mutation in the C-terminal GFP that favours higher expression (Response Figure 1). Furthermore, in the revised manuscript, we include further data for a distinct Monobody, Mb26, which expresses at levels comparable to Mb32 and behaves equivalently to Mb27 (Supplementary Figure 2a, d, e-g). Our data with recombinant proteins indicate that Mb26 competes with RIPK3 for the binding site on MLKL, analogously to Mb27, confirming a common mode of action (Supplementary Figure 3). We have included Mb26 in our revised manuscript, because we feel having a second, distinct Monobody that behaves analogously to Mb27 strengthens our observations and excludes the possibility that our findings with Mb27 are spurious or artefactual. We have included additional text reporting these observations on pages 5 and 6.

2. In Figure 1a and S3, the author analyzed the Mb27-MLKL binding at the time-point 3 hours after necroptosis induction. The Figure S4 showed that the necroptosis began at 4

hours after induction in HT-29 cells. And the previous paper from same group submitting this manuscript have showed that in HT-29 cells the phosphorylation level of MLKL at 6-hour after necroptosis induction was about 5 to 20 folds higher to phosphorylation level of MLKL at 3-hour after induction (Figure 1C from Tanzer, M.C. et al. Biochem J, 2015). Therefore, 3-hour is not enough for these assay, time-course assay containing 6-hour and later time-points is required for Figure 1 and S3. And NSA is recommended if the author wants to test the longer time-point such as 12-hour or 20-hour later from necroptosis induction.

We appreciate the reviewer's careful reading of our study. A more suitable comparison than our earlier Tanzer study is our recent Samson *Nat Commun* 2020 paper, because Tanzer used a less potent stimulus, TSQ, and the HT29 cells were cultured in a less responsive medium, RPMI, which means that timepoints longer than 3 h were necessary to see marked MLKL phosphorylation. However, with TSI as the stimulus, we see near-maximal phosphorylation of MLKL as early as 3 h with cell death from 7.5 h (Samson 2020). Accordingly, the earlier timepoint used here corresponds with instigation of necroptosis as per Samson 2020, but without the death that accompanies later timepoints (7.5 h). However, we agree it is of interest to look nearer to the time of death, and therefore have repeated the experiment with 7.5 h TSI-stimulation of HT29 cells and corresponding Mb27 IP. We have included these data as Figure 1d and accompanying description on page 6, lines 140-147. We describe that, unlike at 3h, we observe RIPK3 bound to MLKL in these Mb27 IPs at 7.5h TSI stimulation. We attribute this phenomenon to the assembly of MLKL oligomers at the necrosome into clusters, as we have described in Samson 2020, where pMLKL are retained in an MLKL oligomer but have disengaged from RIPK3 to expose the Mb27 binding site. We have also commented on the specificity of the pRIPK3 antibody to avoid any confusion amongst the readership; the pRIPK3 antibody detects unmodified RIPK3 also, so we have commented on the upper band serving as the key readout for RIPK3 phosphorylation.

3. The author claimed disengagement of phosphor-MLKL (pMLKL) from RIPK3 can be a checkpoint for MLKL activation and cell necroptosis (in Figure 5). The checkpoint usually means a very important event in biological process, such as the checkpoints in cell cycle.

We take the reviewer's point and have added additional text to ensure our meaning is clear on page 15, line 405. We have defined checkpoint as a hallmark event in pathway chronology that could be considered a regulatory step. In light of the reviewer's comment, we have amended the Abstract to "key regulatory step" rather than "checkpoint".

It indicated disengagement of pMLKL from RIPK3 is necessary for MLKL function. But current data have just shown there was free phosphor-MLKL in cytosol and not in complex with RIP3. It may come from the unphosphorylated MLKL competition of the binding of RIP3 to release the pMLKL. And it may come from the PPM1b removing the pS227 of RIP3 in the complex to decrease RIP3-MLKL binding affinity to release the pMLKL. The author should give more evidences to support the model in Figure 5 that disengagement of pMLKL from RIPK3 is a checkpoint step for MLKL activation and preceding necroptosis. Or it just plays an important (or a small, or no) role in preceding necroptosis.

The reviewer raises some excellent considerations and we thank them for prompting further investigation. We have previously reported that the T357/S358 phosphomimetic MLKL mutant ("TSEE") is a loss of function mutant owing to reduced affinity for RIPK3 (Petrie *et al.*, *Nature Commun* 2018). Here, we have taken the opportunity to generate further evidence

for the phosphorylation of MLKL serving as a trigger for RIPK3 disengagement. We have included two new lines of evidence: one in cells and another using recombinant proteins.

In cells: we expressed FLAG-RIPK3 in *RIPK3*^{-/-} HT29 cells and performed FLAG IPs in the absence of or following necroptotic stimulation. While we saw MLKL bound to RIPK3 in the absence or presence of necroptotic stimuli, crucially, pMLKL was not detected in the RIPK3 IP. These data argue for the idea that MLKL phosphorylation, rather than RIPK3 dephosphorylation, induces MLKL disengagement from RIPK3. We present these data as revised Figure 1e, with text describing these findings on page 6, line 149-154, of the revised manuscript.

Recombinant proteins: we co-expressed untagged human RIPK3 kinase domain and His-tagged MLKL pseudokinase domain in insect cells and examined what cues lead to the complex dissolving. We incubated the complex with Mg²⁺, AMPPNP, or ATP in the presence or absence of Mg²⁺. As evident from these data (presented as Figure 1f in the revised manuscript), only ATP/Mg²⁺ led to dissociation of the MLKL and RIPK3 complex, indicating that it is RIPK3's kinase activity, rather than nucleotide or Mg²⁺ binding, that cause MLKL to break from RIPK3. We discuss these data on page 6 and 7, lines 154-165. Coupled with our earlier finding that the phosphomimetic T357E/S358E MLKL mutant no longer signals for necroptosis (Petrie *et al.*, *Nature Commun* 2018), our new data provide compelling support for the idea that MLKL phosphorylation by RIPK3 is the trigger for dissociation.

4. In figure S1b, the author showed the K_d of Mb27 binding to MLKL. I recommend the author may test K_d of RIP3-MLKL binding in vitro. If the K_d results show RIP3-MLKL binding is stronger than Mb27-MLKL, it may explain why Mb27 do not bind MLKL when RIP3 is present before necroptosis induction.

While we would very much like to examine the affinity of RIPK3 for MLKL, testing the binding of RIPK3 to MLKL *in vitro* is not technically feasible, possibly because it requires a very specific repertoire of PTMs to enable binding of the two proteins when mixed, or possibly owing to non-ideal behaviour (thermal instability) of isolated RIPK3. Our earlier examination of RIPK3:MLKL binding allowed us to make qualitative comparisons between wild-type and mutant MLKL binding to RIPK3 (e.g. Petrie *et al.*, *Nature Commun* 2018 and *Cell Rep* 2019), but did not allow us to perform quantitative analysis owing to incomplete dissociation of the complexes and poor fits to binding models in surface plasmon resonance experiments.

As a result, to study the human complex, we have elected to study recombinant purified human MLKL:RIPK3 complex that we prepared from insect cells as a stable complex using an approach analogous to that taken by Xie *et al.* (*Cell Rep* 2013) for the mouse RIPK3:MLKL complex. This complex only formed when the proteins were co-expressed, which precludes the biophysical study of RIPK3-MLKL interaction using proteins prepared separately. We then examined whether purified human RIPK3:MLKL complexes could bind His-tagged Mb27, Mb32 and an additional Monobody now described in the manuscript, Mb26, in HisTrap mixing/pulldown experiments. As shown in Supplementary Figure 3, and described on pages 5-6, lines 124-135, His-Mb32 could bind the MLKL:RIPK3 complex without the proteins dissociating, whereas His-Mb27 and Mb26 bound only MLKL. As anticipated, a control Monobody, His-Mb33, which we previously reported to bind the 4HB domain (a domain absent from these preparations), does not pulldown MLKL. These data

unambiguously demonstrate that RIPK3 binding to MLKL occludes the Mb27 (and Mb26) binding site, and dissociation of the complex is required for Mb26/Mb27 to bind MLKL.

5. MLKL conformation is different in Mb27-MLKL and Mb32-MLKL crystal structure. But the author mentioned that Mb32 may bind “both the open and closed forms of MLKL” (in line 185 of the manuscript). Is it possible that Mb27 also bind both the open and closed forms of MLKL?

We thank the reviewer for prompting clarification of this important point. Based on the binding interface centred on the pseudoactive site cleft, it is unlikely that Mb27 recognises the open form of MLKL in which the activation loop forms a helix. We now illustrate this by including a superimposition of the Mb27 complex structure with the open form of MLKL crystallised in complex with Mb32 in Supplementary Figure 5b, and we have included accompanying text on lines 233-238 of page 9.

If Mb27 only binds to closed forms of MLKL. It should figure out why in RIP3-KO cells, there is enough closed forms of MLKL to bind Mb27, while with RIP3 expression the closed forms of free MLKL are gone. Did it mean most of MLKL are in complex of RIP3 in untreated cells? The author is recommended to use enough beads containing RIP3 antibody to pull-down RIP3-MLKL complex, and see how much MLKL is in complex with RIP3 and how much MLKL is left being free in cytosol before necroptosis induction. And the free MLKL in cytosol and can be used to test Mb27-binding.

We completely agree with the reviewer’s interpretation of our data, namely that RIPK3 binding to MLKL under basal conditions precludes Mb27 binding to MLKL, while RIPK3-deficiency frees MLKL to interconvert to the closed form that is recognised by Mb27. We agree that the simplest interpretation of our data is that MLKL is indeed in complex with RIPK3 in untreated cells.

Regrettably, the experiment the reviewer has proposed is not trivial; reagents for specific (and complete) IP of endogenous human RIPK3 are currently unsuitable (an extension of the IF in our recent report, Samson *et al.*, *BioRxiv*; and in press at *CDD*) and preclude performing such an analysis in an unambiguous manner. Using an alternative approach, co-expression and purification of the recombinant human MLKL pseudokinase and RIPK3 kinase domain complex from insect cells, however, provides unambiguous proof that co-expressed RIPK3 and MLKL can form stoichiometric complexes (now shown in Figure 1f). We feel, by extension, this provides additional evidence for co-translation of MLKL and RIPK3 promoting their pre-engagement under basal conditions, which we now emphasise in the revised manuscript (page 14, lines 387-394).

We feel there is a simple explanation for why Mb27, which specifically binds the closed form of MLKL (as illustrated in the new Supplementary Figure 5b), is able to recognise MLKL in *RIPK3*^{-/-} HT29 cells. In the absence of RIPK3, MLKL can interconvert from the open, inactive form to the closed active-like conformation, much like what has been proposed for conventional kinases like Aurora A and EGFR, and other pseudokinases, like Ror1 (as per Sheetz *et al.*, *Mol Cell* 2020). This thinking is supported by additional molecular dynamics simulations, which we have included as the revised Figure 3j. Firstly, using an enhanced sampling approach, we observed that the open and closed forms of MLKL represent two energy minima, which accounts for why each were observed in crystal structures. While RIPK3-binding is likely to stabilize the open conformation of MLKL, our simulations

indicate that MLKL phosphorylation lowers the energy barrier to interconversion, which accounts for why the closed form can form post-necroptotic stimulation/RIPK3-mediated MLKL phosphorylation, which leads to Mb27 recognition the closed, phosphorylated form of MLKL. We have included these data as revised Figure 3j with accompanying description on pages 11, lines 287-298 and Discussion on page 15, lines 416-419.

6. The title “Conformational interconversion of MLKL and disengagement from RIPK3 precedes cell death by necroptosis” seems to indicate the conformational interconversion contribute to MLKL disengagement from RIPK3. If the answer is yes, It is interesting to figure out how open to close form transition contributes MLKL disengagement from RIPK3. Q356A mutant disrupts an important salt-bridge in open form of MLKL. So that, it is supposed to have more closed form of MLKL than wild-type of MLKL. It is recommended to test the binding affinity (Kd) in vitro of wild-type and Q356A with RIP3 and Mb27. If Mb27 only binds with closed form of MLKL, and open to close form transition will weaken the RIP3-MLKL binding and induce disengagement of MLKL from RIP3. It may get a result that the binding to RIP3: WT > Q356A ; and the binding to Mb27: WT < Q356A. Otherwise, conformational interconversion of MLKL and disengagement from RIPK3 may be unrelated events.

These are good ideas, but exceedingly difficult to test experimentally. The reviewer is proposing that Q356A mutation would favour the closed form of MLKL and E250A the open form. In the absence of structures of each mutant, we cannot say with any certainty whether these mutations will favour one conformation over the other. Additionally, our MD data indicate the open and closed forms of MLKL exist in an equilibrium, which can be skewed by RIPK3 binding (open form) and phosphorylation (closed form), so we do not expect introduction of these mutations to promote solely one conformer. Nonetheless, using the approach of co-expression in insect cells described above, we prepared recombinant protein complexes of Q356A or E250A human MLKL pseudokinase domain and RIPK3 kinase domain to examine whether these mutations can disrupt complex assembly. Our data (shown in Response Figure 2) illustrate that each MLKL mutant can still complex with RIPK3, although the nature of the complexes differs between the mutants.

[REDACTED]

[This figure has been redacted]

7. E250-K230 salt-bridge is conserved in human, rat horse MLKL and Mb27-MLKL closed form structure. It is interesting to test if E250 mutation has effect on necroptosis proceeding, RIP3-binding and Mb27 binding.

We thank the reviewer for this suggestion, which is very much the complement of the suggestion in point 6 above. We introduced E250A human MLKL into *MLKL*^{-/-} HT29 cells and examined whether this mutant can reconstitute cell death signaling. These data are presented as Supplementary Fig. 2k and are described on page 10, lines 264-268, and show that E250A MLKL can reconstitute signaling in MLKL-deficient cells. We conclude that while the K230:Q356 and K230:E250 interactions are hallmarks of the open and closed conformations, respectively, other factors must contribute to MLKL interconversion.

8. In figure 4 and S4, the author showed alanine mutation in the site R292 in the Mb27 binding epitope prevent necroptosis. Did it disrupt or enhance the binding to Mb27? How about other mutations in Mb27 binding epitope? Mb27 expression did not affect necroptosis proceeding (Figure S2b). So, how R292A prevent necroptosis? Did R292A function to prevent open to closed form transition and disengagement from RIP3? The current cell death results seem not contribute to the model of MLKL in Figure 5. The author should give more related data of the R292A and other mutants for MLKL conformational transition and disengagement from RIP3. Otherwise, these cell death results of MLKL mutants were only confusing but not contributed to the model in Figure 5.

We thank the reviewer for prompting further discussion on these data and inclusion of further experiments. Our model for how MLKL R292 contributes to necroptosis is that it mediates important interactions within the C-lobe of the MLKL pseudokinase domain. We have now highlighted these interactions with a structure cartoon as the revised Fig. 4f. The goal of our

mutational experiments was to identify if further residues in the Mb27 epitope contribute to RIPK3 interaction. Prompted by the reviewer's comments we have now included pMLKL immunoblot analysis, where we use MLKL phosphorylation as a barometer for the propensity of the mutant to interact with RIPK3. We observed R292A MLKL phosphorylation indicating that this mutant retains RIPK3-binding, which was unexpected considering the lack of cell death by R292A MLKL (Figure 4g). We also tested whether R292A MLKL can bind Mb27 by IP, as suggested by the reviewer. As shown in Suppl. Fig. 7b, Mb27 did not pulldown R292A MLKL in the presence or absence of TSI stimulation, but did IP wild-type MLKL post-TSI stimulation.

Another important addition during our revisions is an analysis of the phosphomimetic MLKL mutant, T374D. Previous studies have identified this site as phosphorylated in a cell cycle dependent manner (Daub *et al.*, Dephoure *et al.*). We observed that this mutation led to loss of necroptotic signaling and compromised MLKL phosphorylation (new Figure 4g), indicating a repulsive interaction and abrogation of the RIPK3 interaction. We feel these data add to our understanding and further the value of the revised Figure 5. These data are presented in Figure 4d, e, g and Supp. Figure 6o, Supp. Fig. 7a and b, and are described on page 13, lines 356-368. We present data in Supp. Fig. 7b that show that, like R292A, T374D MLKL did not detectably bind Mb27, in addition to the loss of RIPK3 binding by T374D MLKL.

Reviewer #2 (Remarks to the Author):

Murphy et al. have come up with interesting facts in their study aimed at investigating the conformational transformations observed in pseudokinase domain of MLKL after phosphorylation by RIPK3. This is an interesting study as post-phosphorylation, MLKL undergoes oligomerization to induce cellular necroptosis. The authors have used two monodies to investigate the conformational transformations observed by MLKL after phosphorylation by RIPK3. For that purpose they developed monobodies that specifically target pseudokinase domain of MLKL Mb27 and Mb32 (with KD values of 75 nM and 37.1 nM respectively) The authors have successfully shown that whereas MB32 complexed with MLKL pseudokinase domain, as illustrated by their immunoprecipitation assays, the MB27 immunoprecipitated only with phosphorylated MLKL by RIPK3. These assays showed that necroptotic stimuli induce structural transformations which lead to disengagement of MLKL from RIPK3 consequently presenting the Mb27 binding site. Therefore phosphorylation of MLKL by RIPK3 and its disengagement is pivotal for its role in necroptosis. To investigate the conformational transformations exhibited by MLKL pseudokinase domain, the authors also co-crystallized MB27 and MB32 with MLKL pseudokinase. The authors have observed two distinct open and closed MLKL conformations in these complexes which are mediated by the movement of MLKL activation loop across alpha-c helix. The authors also successfully showed that whereas Mb32 complex structure exhibits open conformation, MB27 binds with pro-necroptotic closed form of MLKL. With aforementioned experiments and analyses, the authors have successfully addressed their questions aiming to investigate structural and functional transformations exhibited by MLKL pseudokinase and role of RIPK3 in inducing these transformation in cellular necroptosis. The article is well written and experiments have been clearly presented and conclusions have been logically drawn, therefore I recommend the article for publication.

We are grateful to Reviewer 2 for their generous appraisal of our study. We thank them for their positivity.

Reviewer #3 (Remarks to the Author):

The manuscript entitled "Conformational interconversion of MLKL and disengagement from RIPK3 precedes cell death by necroptosis" is generally well-written. The authors do a good job of highlighting the novelty and importance of their findings, as well as supporting their conclusions with clearly presented data. The study presents the first experimental evidence that human MLKL, the terminal effector in necroptotic signaling, behaves as a conformationally regulated switch protein. The conclusions rely heavily upon the observed differential binding behaviors of two distinct Monobodies to MLKL, and the structural differences discerned in two separate co-crystal structures.

The two most important advantages of this study are: 1) the demonstration of different conformations of human MLKL, whereas previously only structures from other species demonstrated this difference and 2) data from cells which support that different conformations of MLKL are represented in different signaling states.

The two crystal structures are of sufficiently high quality for publication, but inclusion of a few additional statistics and figures are requested.

A drawback of the study in its current form is how binding of Mb27 occludes RIPK3 binding to MLKL. The authors provide structure-based mapping that suggest the binding sites overlap somewhat, yet more evidence would make a stronger argument. For example, direct evidence of competition between Mb27 and RIPK3 would support an overlapping binding site.

We completely agree with the reviewer. We have now generated data that show mutual exclusivity of RIPK3 and Mb27 binding to MLKL, which we have included as Supplementary Figure 3 in the revised manuscript and describe the data on pages 5 and 6, lines 124-135. These experiments examine whether His-tagged Monobodies can bind to the recombinant human RIPK3 kinase domain:MLKL pseudokinase domain heterocomplex co-expressed and purified from insect cells. These data show unambiguously that His-Mb27 (and the new Monobody included in revision, Mb26) can only bind MLKL when RIPK3 dissociates, while His-Mb32 can bind to MLKL in complex with RIPK3. These data provide convincing support for the idea that RIPK3 and Mb27 (plus Mb26) bind to an overlapping site on MLKL. We are grateful to the reviewer for prompting further investigation of this important point.

Alternatively, assessment of the R292A mutant MLKL's ability to bind to RIPK3 and Mb27 would add value to the conclusion. Also, while the other alanine mutant MLKL constructs did not show changes in cell death kinetics, whether they had nevertheless demonstrated diminished binding to RIPK3 and/or Mb27 was not assessed. It seems possible that binding to these mutants could be reduced but the downstream effect on cell death is not noticeably changed. While performing these experiments may be outside the scope of the current study, at the very least, a more thorough discussion of the overlap between the presumed RIPK3 and Mb27 binding sites would benefit the strength of the argument.

We agree and have now added MLKL phosphorylation data as a readout of RIPK3 engagement (as revised Figure 4g). We have included additional discussion of our reasoning in the revised text to reflect our thinking that cell death kinetics of mutants akin to wild-type MLKL are indicative of comparable RIPK3 engagement (pages 12, lines 338-340).

We have provided further description of why R292A mutation leads to defective MLKL function (pages 12-13, lines 342-344; and new Fig. 4e) and, interestingly, observed that MLKL phosphorylation was retained in cells expressing R292A MLKL following TSI treatment (Fig. 4g). We have also included additional studies of the T374D MLKL mutant, a substitution designed to mimic a phosphorylation event reported to occur in a cell cycle dependent manner (by Daub *et al.* and Dephore *et al.*). These studies revealed defective RIPK3 engagement (as read out by abrogated MLKL phosphorylation; Fig. 4g) and cell death when introduced into *MLKL*^{-/-} HT29 cells (Figure 4e). Furthermore, like R292A, T374D MLKL does not detectably bind Mb27 (Suppl. Fig. 7b), and together these data support the idea that the RIPK3 and Mb27 binding sites overlap. We have elaborated on these data validating this overlap as per the reviewer's request in the revised text (page 13, lines 363-368).

The following points are also related:

- How similar are the human and mouse MLKL sequences at the RIPK3 binding site? When discussing how the human MLKL/RIPK3 interface was modelled using the mouse structure, it would be informative to include the %identity/similarity between mouse and human at that interface.

We thank the reviewer for this suggestion. We have included additional panels as revised Figure 4a-b to show identical residues in human and mouse MLKL in a sequence alignment and painted on to the mouse MLKL:RIPK3 complex structure cartoon, and we have included additional text describing conservation (pages 11-12, lines 311-314).

- What is the rmsd upon superposition of human MLKL from Mb27-bound structure with mouse MLKL from RIPK3-bound structure?

This is another good point. We have now described this in the revised text (page 12, lines 315-6); the RMSD is 0.735 Å.

Other points and questions:

- In RIPK3 knockout cells, Mb27 is able to bind MLKL, suggesting that MLKL can adopt the closed/active conformation in the absence of phosphorylation. Or alternatively, does another kinase phosphorylate MLKL to achieve this closed conformation in cells lacking RIPK3? Or does MLKL exist in this closed conformation in the basal state (in equilibrium with the open conformation)? Have the authors probed the Mb27 IP/MLKL coIP in RIPK3 knockout cells for phospho-MLKL?

These are all excellent questions. We have now included new MD simulations to address the question of whether the two conformations might co-exist (Figure 3j). Indeed, our calculations indicate the existence of two energy minima corresponding to the two different states we crystallized, and that MLKL phosphorylation promotes interconversion from open to closed form, which our biochemical data also corresponds with dissociation from RIPK3.

To date, only RIPK3 is known to phosphorylate MLKL to promote necroptotic death. In light of the reviewer's query, we examined whether MLKL was phosphorylated in RIPK3-deficient cells (included in revised Figure 1b; and described on page 5, lines 119-121), but did not detect pMLKL in the absence of RIPK3. The simplest interpretation of our data is that RIPK3 binding to MLKL occludes Mb27 interaction and, based on the reviewer's query, we have elaborated on this line of thought on page 15, lines 429-434.

- Lines 140 and 172, please remove "supplementary"

Removed with thanks.

- The comparisons of the different conformations of the pseudokinase domain across the structures would benefit from a simple superposition of the structures from Figures 3a-b-c as a simple C α trace, preferably by superposing the C-lobes in order to better show the differences in N-lobe orientation with respect to the C-lobe.

This is a good idea; we thank the reviewer for the suggestion. Figure 3 now updated with an additional panel (Fig. 3g) to show C α superpositions.

- Please include B factor information in Table 1 for monobodies and MLKL separately. In addition, it would also be of value to include the B factors for the N- and C-lobes of the pseudokinase domains separately.

Now added to Table 1. We thank the reviewer for this suggestion.

- Could the authors consider including a representative region of electron density, either 2Fo-Fc and/or an omit map, for particular regions of interest in each structure? For example: the area including activation loop/helix and the hydrogen bond between K230 and Q356 from the Mb32 structure, and between K230 and E250 in the Mb27 structure; alternatively, the electron density at the interface of the pseudokinase domain and Monobodies would demonstrate the quality of the maps.

We have now included a new figure, Supplemental Figure 5, to illustrate the quality of the maps, as per the reviewer's suggestion.

- Does the crystal packing reveal any other kinase/Monobody pair that could represent another interface that might exist in solution?

This is a good point, and important for us to clarify. We have now included interface analysis from PDBePISA that indicates that the pseudokinase:Mb pairs that we have reported are the most plausible. We have updated the text to ensure this is clear (pages 8, lines 196-199; page 9, lines 225-228). Coupled with the inclusion of additional data demonstrating that Mb27 does not bind to the R292A and T374D MLKL mutants, we feel the Mb27 binding site we have depicted is now additionally unambiguously supported by mutational data.

- The Ramachandran distribution is a bit problematic, with less than 98% in the favored region in both structures. At these modest resolutions, the geometry could be improved some through tighter restraints during refinement, though would likely not change the overall conclusions of the study.

We agree with the reviewer and thank them for the suggestion. We have now re-refined our structures, deposited updated coordinates, redrawn the structure figures with the improved geometry and we have updated Table 1 accordingly.

- The authors should briefly elaborate in their Methods section on the search models used for molecular replacement. Was the molecular replacement search model for MLKL a single kinase domain, or was it split into N- and C-lobes and searched individually? At these modest resolutions, and with large conformational differences between structures, a clearer picture of how the solutions were obtained could be very helpful for a reader.

We thank the reviewer for this suggestion; we have added more details to the methods (page 22, lines 617-621).

REVIEWER COMMENTS

Reviewer #1 (Remarks to the Author):

Comments for Author

The authors have addressed most of my queries, but there are still some points need to be further clarified.

1. Usually, co-expression a kinase and substrate protein in insect cell will cause strong substrate-phosphorylation. So, the Figure 1f of the new manuscript lacks the necessary results such as p-RIP3 and p-MLKL. The author should test the input lysis and the Ni²⁺ eluates with not only RIP3 and MLKL but also p-RIP3 and p-MLKL.
2. In the new manuscript, the author showed two Mb27- and Mb32- IP results: One is from the samples at 3 h after TSI treatment(Figure 1a), the other is at 7.5 h after TSI treatment(Figure 1d). As the author have mentioned in the rebuttal letter, they “see near-maximal phosphorylation of MLKL as early as 3 h with cell death from 7.5 h” . So that the IP results from samples of 3-hour TSI treatment and 7.5-hour TSI treatment should be similar. But I found these results differ. RIP3 is in the Mb27-IP at TSI-7.5-hour sample but not TSI-3-hour sample. Besides that, the RIP3 signal is much stronger in the Mb32-IP at TSI-7.5-hour sample than TSI-3-hour sample. It may suggest these differences were due to the different exposure time? The author should put the 3-hour IP and 7.5-hour IP in the same gel to avoid this possibility.
3. It is still confusing how R292A prevent necroptosis? The author found it did not due to deficiency of the RIP3 binding and phosphorylation by RIP3. I suggested author gave some specific reasons or possibilities in the discussion part.

Reviewer #3 (Remarks to the Author):

The authors have directly addressed my concerns raised in the first review. The newly refined crystal structures have both improved and sufficient statistical quality, and the inclusion of additional structure-related information in the manuscript reflects this. I now recommend without reservation this revised manuscript for publication.

We thank the reviewers for their careful consideration of our revised manuscript. We thank Reviewer 3 for recommending publication of the work; we hope that the presentation of additional data shown below in response (black roman text) to reviewer 1's queries (*blue italics*) and the additional text on page 13 of the revised version (shown **highlighted in grey**) now satisfy any outstanding concerns.

REVIEWER COMMENTS

Reviewer #1 (Remarks to the Author):

Comments for Author

The authors have addressed most of my queries, but there are still some points need to be further clarified.

1. Usually, co-expression a kinase and substrate protein in insect cell will cause strong substrate-phosphorylation. So, the Figure 1f of the new manuscript lacks the necessary results such as p-RIP3 and p-MLKL. The author should test the input lysis and the Ni²⁺ eluates with not only RIP3 and MLKL but also p-RIP3 and p-MLKL.

This is absolutely the case; it is typical for substrates to be phosphorylated by their cognate kinase when co-expressed as recombinant proteins in insect cells as the reviewer rightly pointed out. This is exactly what we see when probed for pMLKL (Response Figure 1).

There are a number of important points to consider here, which make this system difficult to interpret unambiguously based on the information to hand. We have not modified the previous version of the manuscript in light of the reviewer's query so as to not overcomplicate our interpretation or confuse the reader, but we offer the following reasoning:

- Dissolution of the RIPK3:MLKL complex (Figure 1f) relies on the presence of both ATP and Mg²⁺. This is unambiguous; the kinase activity of RIPK3 is absolutely required for the complex to come apart. Because this is unambiguous, we have emphasised this when interpreting and describing these data.
- As shown in Response Figure 1, pMLKL is indeed present in the purified complex and persists after ATP/Mg²⁺ treatment. What was surprising is that the pMLKL signal did not increase following the kinase reaction, raising the possibility that detection by western blot may not reflect the increase owing to saturation of the signal or by occlusion of pMLKL antibody binding due to another phosphorylation event introduced by RIPK3.
- The available commercial antibodies only reliably detect phosphorylation of S358 within the human MLKL activation loop. One possibility, which is a widely held view

in the field (based on L Sun *et al.*, *Cell* 2012), is that it is phosphorylation of the adjacent residue, T357, is also required for dissociation from RIPK3. Our recent findings indicate that the commonly used pT357 MLKL antibody is not suitable to detect this event (Samson *et al.*, *CDD* 2021), which means this possibility remains an open question. Furthermore, as mentioned above, it is conceivable that T357 phosphorylation could occlude binding to pS358. This remains far from clear at present; our ongoing studies are geared towards understanding if phosphorylation of residues other than S358 in MLKL is required for dissociation from RIPK3. We feel such a study represents a discrete and substantial body of work beyond the scope of the present manuscript. We expect to report this in a separate manuscript when we have more answers.

2. In the new manuscript, the author showed two Mb27- and Mb32- IP results: One is from the samples at 3 h after TSI treatment(Figure 1a), the other is at 7.5 h after TSI treatment(Figure 1d). As the author have mentioned in the rebuttal letter, they “see near-maximal phosphorylation of MLKL as early as 3 h with cell death from 7.5 h” . So that the IP results from samples of 3-hour TSI treatment and 7.5-hour TSI treatment should be similar. But I found these results differ. RIP3 is in the Mb27-IP at TSI-7.5-hour sample but not TSI-3-hour sample.

We thank the reviewer for seeking clarification of this point. We completely agree with the reviewer’s interpretation of these data. In the previous version of our manuscript (now page 6, lines 144-147 of the current version), we discussed the point that we only see RIPK3 coming down in Mb27 IPs at 7.5 h post-TSI because we believe pMLKL is engaged in oligomeric clusters at 7.5 h that are not in full flourish at 3h post-TSI (also as described in Samson *et al.*, *Nat Comm* 2020). It is important to remember that the Monobodies are immunoprecipitating MLKL, rather than RIPK3; any RIPK3 in the complex is present entirely because of its association with MLKL. Because it is not RIPK3 being IP’d by the Monobodies directly, it is not surprising that the bound RIPK3 differs depending on the time post-TSI and whether pMLKL is engaged in complex with RIPK3 at that timepoint. The finding that RIPK3 only co-IP’d with Mb27 at later time points, but not 3h, is further supported by the blots included below as Response Figure 2. The presence of RIPK3 in Mb27 co-IPs at later time points, but not 3h post-TSI or earlier, was consistent throughout many repeats for both Mb27 and a second Monobody with similar binding characteristics, Mb26.

Besides that, the RIP3 signal is much stronger in the Mb32-IP at TSI-7.5-hour sample than TSI-3-hour sample. It may suggest these differences were due to the different exposure time? The author should put the 3-hour IP and 7.5-hour IP in the same gel to avoid this possibility.

We have included a timecourse experiment below for the reviewer’s benefit (Response Figure 2), which includes 3h and 7.5h post-TSI IPs side-by-side on the same gel. As a result, any differences in co-IP’d RIPK3 observed are not attributable to differences in exposure times.

**Response
Figure 2**

3. It is still confusing how R292A prevent necroptosis? The author found it did not due to deficiency of the RIP3 binding and phosphorylation by RIP3. I suggested author gave some specific reasons or possibilities in the discussion part.

We sought to offer the most conservative interpretation of these data during our last round of revision. In light of the reviewer's query, however, we have now speculated further on why this may be (page 13 of the revised manuscript; lines 346-7 and 354-6). Our thinking is that the reduced expression may reflect the instability of the MLKL mutant that would be expected from perturbation of structural interactions. We also suggest a possible role downstream of membrane association, where activated MLKL protomers may exhibit deficits in higher order assembly into hotspots.

Reviewer #3 (Remarks to the Author):

The authors have directly addressed my concerns raised in the first review. The newly refined crystal structures have both improved and sufficient statistical quality, and the inclusion of additional structure-related information in the manuscript reflects this. I now recommend without reservation this revised manuscript for publication.

Again, we thank Reviewer 3 for their positivity and recommendation.

REVIEWER COMMENTS

Reviewer #1 (Remarks to the Author):

I have no more question for the revised manuscript. But I think it should be careful to the result that Mb27 could pull-down more the RIP3 complexed with MLKL at later timepoint after TSI treatment (Response Figure 2). It has some conflict with one of the conclusions of this manuscript: RIP3-phosphorylated MLKL tends to dissociate from the RIP3-MLKL complex, which suggested the more dissociated MLKL at the later timepoint, the less MLKL-complexed RIP3 should be pulled down by Mb27.

Reviewer #1 (Remarks to the Author):

I have no more question for the revised manuscript. But I think it should be careful to the result that Mb27 could pull-down more the RIP3 complexed with MLKL at later timepoint after TSI treatment (Response Figure 2). It has some conflict with one of the conclusions of this manuscript: RIP3-phosphorylated MLKL tends to dissociate from the RIP3-MLKL complex, which suggested the more dissociated MLKL at the later timepoint, the less MLKL-complexed RIP3 should be pulled down by Mb27.

We have addressed this over the course of our review conversation with the reviewer. We have provided an explanation based on our earlier findings (in lines 144-147 of the main text). It is possible the reviewer does not agree with our interpretation, however we feel that by presenting the data in entirety, including in the transparent peer review document, readers will be able to draw their own conclusions.